# Cache Me If You Must: Adaptive Key-Value Quantization for Large Language Models

**Alina Shutova** [* 1 2]  **Vladimir Malinovskii** [* 2 1]  **Vage Egiazarian** [* 3]  **Denis Kuznedelev** [2]  **Denis Mazur** [4 5]
**Nikita Surkov** [6]  **Ivan Ermakov** [5 2]  **Dan Alistarh** [3]

## Abstract

Efficient real-world deployments of large language models (LLMs) rely on Key-Value (KV) caching for processing and generating long outputs, reducing the need for repetitive computation. For large contexts, Key-Value caches can take up tens of gigabytes of device memory, as they store vector representations for each token and layer. Recent work has shown that the cached vectors can be compressed through quantization, pruning or merging, but these techniques often compromise quality towards higher compression rates. In this work, we aim to improve Key & Value compression by exploiting two observations: 1) the inherent dependencies between keys and values across different layers, and 2) the existence of high-compression methods for internal network states (e.g. attention Keys & Values). We propose AQUA-KV, an adaptive quantization for Key-Value caches that relies on compact adapters to exploit existing dependencies between Keys and Values, and aims to "optimally" compress the information that cannot be predicted. AQUA-KV significantly improves compression rates, while maintaining high accuracy on state-of-the-art LLM families. On Llama 3.2 LLMs, we achieve near-lossless inference at 2-2.5 bits per value with under 1% relative error in perplexity and LongBench scores. AQUA-KV is one-shot, simple, and efficient: it can be calibrated on a single GPU within 1-6 hours, even for 70B models.

## 1. Introduction

Large Language Models (LLMs) are revolutionizing natural language processing, but come with major computational costs, in particular due to the input-quadratic complexity of

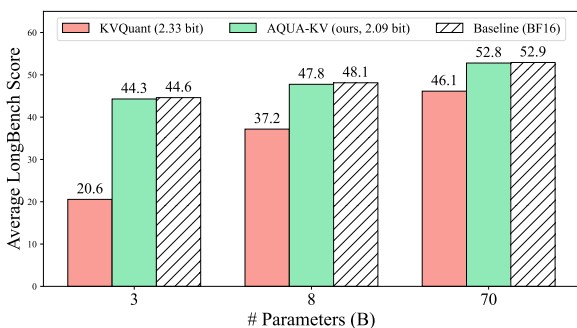

*Figure 1.* Comparison of AQUA-KV to alternative Key-Value Cache compression methods for Llama 3.x models in terms of average LongBench score on 14 english tasks (see Section 4).

attention-based Transformer models (Vaswani, 2017). To achieve faster inference and avoid wasteful recomputation of attention scores during autoregressive generation, *KV caching* is typically employed, where keys and values are saved for later use. Unfortunately, KV-caching comes with its own pitfalls: KV caches are large, especially when handling long sequences (Bai et al., 2023; Xiao et al., 2023). Thus, the memory footprint of a full-length Key-Value cache can reach tens of gigabytes of device memory, sometimes more than the model itself[1]. This massive memory consumption increases the cost of deployment, and also slows down inference, as the whole process can become memory-bound for large caches (Hooper et al., 2024).

Previous work has proposed methods to compress KV caches using various methods such as quantization and pruning (Li et al., 2024a), that can significantly reduce the memory footprint of KV caches. Yet, as we increase the degree of compression, e.g. 2 bits per value, existing compression techniques begin to lose significant information, resulting in poor accuracy (Li et al., 2024c).

In this work, we aim to improve KV cache compression by taking advantage of the inherent structure and dependencies in the cache tensors. Specifically, we analyze the Key-Value cache behavior for state-of-the-art LLMs and find several

---

[*]Equal contribution  [1]HSE University [2]Yandex [3]ISTA [4]SberDevices [5]MIPT [6]T-Bank. Correspondence to: Dan Alistarh <dan.alistarh@ist.ac.at>.

*Proceedings of the 42nd International Conference on Machine Learning*, Vancouver, Canada. PMLR 267, 2025. Copyright 2025 by the author(s).

---

[1]For the popular Llama 3.2 3B model (Dubey et al., 2024) with a maximum context length of $2^{17}$ tokens ($\approx$131K), the 16-bit cache takes up 15 GB per sequence. For Llama 3.1 70B and Qwen 2.5 72B, it is 42.9 GB per sequence.

strong inter-dependencies 1) between cached vectors from adjacent layers, but also 2) between the keys and values within one layer.

Starting from these observations, we formulate a practical compression algorithm that explicitly leverages these inter-dependencies, by training compact linear predictors that capture mutual information between cache components. Further, to offset prediction errors, we use data-free vector quantization to achieve a superior compression-accuracy trade-off for the same bit-width. Our method requires only minimal calibration, is compatible with arbitrary quantization schemes and can be further combined with orthogonal compression techniques such as pruning.

In summary, our contributions are as follows:

1. We analyze the structure of key-value caches in modern LLMs and highlight several sources of mutual information that can be leveraged for compression.

2. We propose AQUA-KV — a novel compression framework that exploits inter- and intra-layer dependencies to improve quantization accuracy. AQUA-KV works in one shot, based on a lightweight calibration procedure, and shows competitive size-accuracy trade-offs. Additionally, it is compatible with arbitrary quantization techniques, and can be combined with additional compression, such as pruning.

3. We validate the effectiveness of AQUA-KV on modern LLM families in terms of both perplexity and zero-shot accuracy on long-range benchmarks, where AQUA-KV significantly improves accuracy across model types and bitwdiths, particularly for 2-bit compression.

4. We test AQUA-KV compatibility with various quantization and pruning schemes, from simple uniform quantization, to modern data-free vector quantizers (Malinovskii et al., 2024b) and hybrid quantization & pruning regimes.

5. We develop a reference implementation for AQUA-KV calibration and inference, which is available online[2].

## 2. Background and Related Work

### 2.1. KV-Cache Compression

So far, the main focus of work on LLM compression has been on the *weight quantization*, e.g. (Frantar et al., 2022; Lin et al., 2023; Tseng et al., 2024a; Egiazarian et al., 2024). Recently, there has been a growing demand for KV-cache compression, especially in tasks requiring long contexts. The dynamic nature of KV-caching poses unique challenges: while for weights it is acceptable to use "slow but accurate" compression such as codebook-based methods (Tseng et al., 2024a; Egiazarian et al., 2024)—given that weights are only decoded at inference time—for the KV-cache, *both*

*compression and decompression speeds matter*, since we are dynamically adding new entries to the cache as well as decoding them at inference time. Another issue is posed by the inherent structure in caches, in particular, the existence of attention sinks (Xiao et al., 2023) and large outlier values (Liu et al., 2024c; Hooper et al., 2024; Liu et al., 2024a), which may not be present in the weights. Next, we detail the main approaches for KV-cache quantization.

KV-Cache quantization approaches can be roughly categorized based on quantization granularity and error handling.

**Quantization Granularity.** Several approaches have been developed with varying quantization granularities. For instance, ZipCache (He et al., 2024) and WKVQuant (Yue et al., 2024) implement channel-separable token-wise quantization, while KVQuant (Hooper et al., 2024) and KIVI (Liu et al., 2024c) employ a hybrid approach, using per-channel quantization for key tensors while applying per-token quantization for value tensors. QJL (Zandieh et al., 2024) introduces a specialized JL transform for key tensors combined with per-token quantization for value tensors. Methods like MiKV (Yang et al., 2024b), QAQ (Dong et al., 2024), and SKVQ (Duanmu et al., 2024) employ variable bit widths to balance accuracy and memory reduction.

**Error Handling.** Among the strategies used to address quantization errors, GEAR (Kang et al., 2024) compensates for errors using a low-rank matrix; to handle outliers, which can significantly impact model performance, IntactKV (Liu et al., 2024b) maintains full precision for outlier values. QuaRot (Ashkboos et al., 2024) transforms weight matrices using Hadamard orthogonal matrices to "smoothen" quantization outliers without affecting model output. Palu (Chang et al., 2024) compresses KV cache through low-rank projection, while ZDC (Zhang & Shen, 2024) aims to eliminate compression overhead through a novel zero-delay compression scheme. Most methods maintain a window of recent historical KV cache in full precision to preserve accuracy.

**Cross-Layer Merging.** An alternative compression approach, which has been relatively less investigated, has been to define layer groups, and keep a single KV-Cache per layer group, reusing the one cache across all layers in the group. KVSharer (Yang et al., 2024c) observes that, surprisingly, sharing the caches that are *most dissimilar*, by the Euclidean distance, performs better than sharing similar ones. (This is identified using a calibration dataset.) Further, MiniCache (Liu et al., 2024a) merges pairs of layers, and stores a common interpolated directional vector and token-wise scalar scales. Additionally, outlier tokens are retained to increase accuracy. During inference, keys and values of the two layers are restored from the shared vector representation using saved scales.

**KV Cache Pruning.** Pruning methods aim to determine unnecessary parts of KV Caches and either evict them en-

---

[2]https://github.com/goodevening13/aquakv

tirely, or offload to cheaper memory (e.g. CPU). Current research focuses on token-level pruning, e.g. determining which tokens should be discarded or offloaded. They can be split into static methods and dynamic methods.

*Static methods* use predefined position heuristics to determine important tokens. An example is Fastgen (Ge et al., 2023), which employs knowledge of attention structure in attention heads, acquired during prefill, to identify one of the tailored attention structures for each head. This knowledge is then used during inference to efficiently evict unnecessary tokens. StreamingLLM (Xiao et al., 2023) notices that initial tokens (attention sinks) and recent tokens consistently exhibit high importance. The method retains several tokens in the beginning and a number of recent tokens, thus maintaining constant-size KV cache, facilitating deployment in memory-constrained scenario.

*Dynamic methods* calculate importances dynamically, typically using runtime information about attention distribution. H2O (Zhang et al., 2023) calculates token importances dynamically during inference using accumulated attention scores. It selects the least important tokens during each forward pass (if the cache is larger than desired) and thus maintains constant cache size. SnapKV (Li et al., 2024b) calculates token importances using attention features; unlike H2O, SnapKV thresholds token importances, allowing prompt compression during prefill.

For our work we choose to conduct experiments using H2O pruning as it is a well-established plug-and-play method. However, we emphasize that our method is orthogonal to pruning and AQUA-KV can be combined with any cache eviction strategy. AQUA-KV integrates both Quantization and Cross-Layer Merging strategies. It can be regarded as more advanced Layer Merging approach via training a small supplementary model, referred to as *predictor*, capturing inter-layer dependencies of KV caches.

**Vector Quantization** recently emerged as a popular option for LLM quantization (Egiazarian et al., 2024; Tseng et al., 2024b; van Baalen et al., 2024) since it allows to jointly quantize multiple individual model dimensions and can lead to state-of-the-art accuracy-vs-compression (Tseng et al., 2024c; Malinovskii et al., 2024a). However, the extremely large computational cost of encoding makes such methods impractical in the context of KV-cache compression.

We resolve this issue, and leverage the power of VQ in an efficient way, by adapting the HIGGS weight quantization technique (Malinovskii et al., 2024b) to KV-Cache compression. HIGGS combines group-wise vector quantization with a Randomized Hadamard Transformation (RHT): the RHT projects the original values onto a "rotated" space, where they will be normally-distributed. In the quantization step, the rotated values are grouped and rounded to the nearest

points on a lattice, which is specifically optimized for accurate quantization of normally-distributed vectors. This allows HIGGS to achieve *fast data-free weight quantization*. The technique requires several modifications to be applied to Key-Value caches, which we detail in Section 4.2. Finally, our approach is conceptually related to Residual Vector Quantization (RVQ) (Gray & Neuhoff, 1998), but using learned predictors instead of standard quantization.

**Linearity between adjacent layers.** Prior work (Razzhigaev et al., 2024) has shown that there is a almost linear relationship between activations in sequential layers in transformer language models due to the presence of skip connections. We leverage this property for the design of KV-cache predictors.

## 3. Method

The core idea of AQUA-KV is to leverage inter-dependencies between consecutive KV-caches to improve compression. **For this, we train compact predictors that "guess" the value of a Key & Value pair using other cache entries, then quantize the residual information that could not be predicted.** This way, we only store the information that cannot be recovered from other sources.

In Section 3.1, we analyze the dependencies between various KV cache components to determine the type of predictors that achieves the best size-accuracy trade-off. In Section 3.2, we formulate a practical one-shot algorithm that fits these predictors for use in KV cache compression. Finally, in Section 3.3 we describe a number of important implementation details for using AQUA-KV in practice.

### 3.1. Analysis of Inter-Layer Dependencies

The efficacy of our approach depends on choosing which types of inter-dependencies to exploit. To make this choice, we analyze the dependencies between cached vectors at different layers or tokens, and between (key & value) vectors within the same layer. Note that we do not expect the values in these vectors to be equal or even numerically close— indeed, a simple examination shows that this is not the case. Instead, we look for consistent dependencies between these components that can be extracted with simple models.

To measure dependencies between vectors at nearby layers, we adopt an approach similar to probing (Alain & Bengio, 2016): we train *linear* "probe" models whose goal is to predict the contents of a particular cache component, e.g. $i$-th layer keys or values, based on inputs from various sources. As potential sources, we consider the previous layer keys and values, adjacent tokens, and different vector types (i.e. using the keys to predict values and vice versa).

We can then measure the relative prediction error for such probes, based on small amounts of calibration data. More

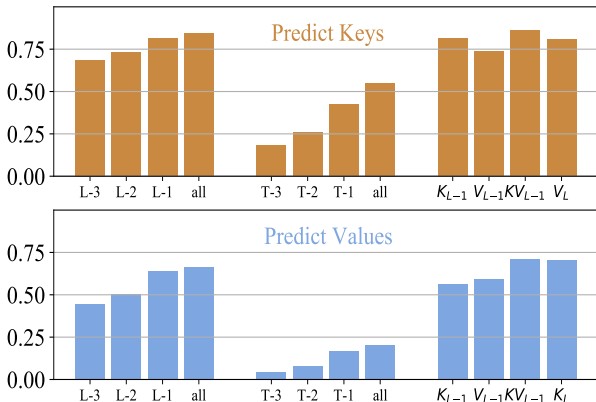

*Figure 2.* Mean Explained Variance Ratios by linear probes from previous blocks (L), tokens (T) and role on Llama-3.2-3B.

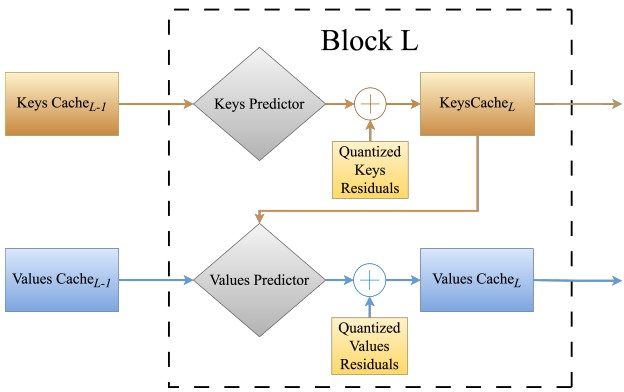

*Figure 3.* An intuitive scheme of the AQUA-KV inference. Only the quantized residuals are saved for each block.

specifically, we measure the explained variance ratio to account for unequal scales of keys and values between layers. Intuitively, if a predictor captures 90% of the variance, it means that the subsequent quantization only needs to capture the remaining 10% of variance. For compression methods that are scale-independent [3](Horváth et al., 2023) this would mean that the resulting quantization will also have roughly 10 times smaller error.

Note that not all predictors will be practical for Key-Value compression. For instance, if a predictor uses a subsequent block or future token KV vectors as inputs for the current ones, it would be difficult to inference the model with such KV cache as it goes against the order in which the blocks execute. Hence, we consider the following:

- **Previous blocks:** same token vectors for -1, -2, -3 blocks;
- **Previous tokens:** same layer, -1, -2, -3 previous tokens;
- **Different role:** using keys to predict values and vice versa.

We report these errors in Figure 2. For comparison, 1-bit and 2-bit quantizers usually explain 0.75 and 0.89 variance, respectively. In other words, they have 0.25 and 0.11 relative quantization errors. Intuitively, if a probe can predict keys/values with the same relative error as the 1- or 2-bit quantizer, it means that we can use 1 less bit for quantization (e.g. 3-bit instead of 4-bit) after the residual with, on average, the same accuracy. While this is not a strict guarantee, we found that it holds well for real-world LLMs, as can be seen in Sections 4.1 and 4.2.

We train linear probes for Llama-3.2-3B Key-Value cache on a sample of RedPajama (Weber et al., 2024) sequences, then evaluate relative error on hold-out sequences from the same source. For readability, the detailed experiment configuration is deferred for Appendix A.

The findings in Figure 2 demonstrate strong dependencies

---

[3]Formally, a scale-independent quantizer $Q(\cdot)$ satisfies $\forall X, \forall \alpha > 0$, $||Q^{-1}(Q(\alpha X)) - \alpha X||_2 = \alpha ||Q^{-1}(Q(X)) - X||_2$. Modern quantizers satisfy this due to the use of scales.

between several cache components. For attention keys, using just one previous layer already achieves errors similar to 2-bit quantization. For values, the dependency on previous layer is also strong: though less accurate than for keys, the previous layer values consistently explains more than half of the variance for the same token. We attribute this strong dependency to the fact that transformer architecture is residual, and therefore, adjacent hidden states are only off by a single transformer layer. Since Key-Value representations are constructed as linear projections of adjacent residual hidden states, they are also interdependent.

More distant layers are also predictive of the current layer, but the dependency quickly deteriorates with the distance. More importantly, there is almost no difference between using multiple past layers and just the previous layer, which allows us to simplify the algorithm. See Appendices B & D for additional exploration and downstream results.

We also observe strong dependencies between keys and values within the same layer. This is also not unexpected, as these vectors are linear down-projections from the same input vector within the attention layer. The reason why there is no additional information between matching keys/values is that in modern LLMs the key/value dimensions are both significantly smaller than the input vector dimension due to Grouped Query Attention (GQA, Ainslie et al. 2023).

In contrast, we found relatively little predictive power from past *tokens*. While *there is* mutual information between adjacent tokens, using a single previous token is not enough to capture this dependency, and using many multiple tokens or a more expressive probe would make the resulting KV cache compression inefficient.

Some of our observations above correlate with the results from Liu et al. (2024a), who were able to "share" some of the cache entries. However, such direct merging is a drastic approach, that can lead to significant accuracy loss. From the perspective of Figure 2, this is because the information recovered by a probe is not enough by itself, never exceeding

the equivalent of a 2-bit quantizer. In contrast, we employ a more fine-grained approach that combines predictors with residual quantization to achieve more accurate compression.

---

**Algorithm 1** AQUA-KV Calibration

---

**Require:** model, data, quantization method $Q$
1: $\mathbf{X} \leftarrow$ model.input_embeddings(data)
2: $\mathbf{K}_{old}, \mathbf{V}_{old} \leftarrow \emptyset$
3: predictors $\leftarrow \{\}$
4: **for** $i = 1, \ldots,$ model.num_layers **do**
5:    block $\leftarrow$ model.transformer_layers[$i$]
6:    $(\mathbf{K}, \mathbf{V}) \leftarrow$ block.get_attention_kv($\mathbf{X}$)
7:    $\mathbf{X} \leftarrow$ block($\mathbf{X}$)
8:    **if** $\mathbf{K}_{old} = \emptyset$ **and** $\mathbf{V}_{old} = \emptyset$ **then**
9:        $\mathbf{K}_{old}, \mathbf{V}_{old} \leftarrow \mathbf{K}, \mathbf{V}$
10:       **continue**
11:   **end if**
12:   # Predict keys from past keys
13:   $f_{key} \leftarrow \arg\min_f \| f(\mathbf{K}_{old}) - \mathbf{K} \|_2^2$
14:   $\mathbf{K}_{residue} \leftarrow Q^{-1}(Q(\mathbf{K} - f_{key}(\mathbf{K}_{old})))$
15:   $\mathbf{K}_{rec} \leftarrow f_{key}(\mathbf{K}_{old}) + \mathbf{K}_{residue}$
16:   # Predict values from past values and current keys
17:   $f_{value} \leftarrow \arg\min_f \| f([\mathbf{V}_{old}; \mathbf{K}_{rec}]) - \mathbf{V} \|_2^2$
18:   $\mathbf{V}_{residue} \leftarrow Q^{-1}(Q(\mathbf{V} - f_{value}([\mathbf{V}_{old}; \mathbf{K}_{rec}])))$
19:   $\mathbf{V}_{rec} \leftarrow f_{value}([\mathbf{V}_{old}; \mathbf{K}_{rec}]) + \mathbf{V}_{residue}$
20:   predictors[i] $= (f_{key}, f_{value})$
21:   $\mathbf{K}_{old}, \mathbf{V}_{old} \leftarrow \mathbf{K}_{rec}, \mathbf{V}_{rec}$
22: **end for**
23: **return** predictors

---

### 3.2. The AQUA-KV Algorithm

Based on the findings from the previous section we design a cache compression algorithm that leverages the structure of Key-Value cache to improve compression.

We choose the following predictor configuration: (1) we use the previous layer keys to predict the subsequent keys; and (2) we use both previous layer values and current layer keys to predict values. Note that we do *not* use the dependency between same layer K and V vectors in the $V_L{\rightarrow}K_L$ direction: this is because we cannot predict in both directions simultaneously during inference, and we found that the values are overall harder to predict (see Figure 2), so we opted to improve value predictor. We also purposefully leave out other possible input sources, such as more distant past layers. While using these layers can slightly improve reduce error, it would also make the predictors themselves larger and more compute-intensive.

The way AQUA-KV trains those predictors is also different from Section 3.1. This is because, in practical KV cache compression, the predictors do not have access to ground-truth past KVs during inference. Instead, they can only use reconstructed (de-quantized) past key and value vectors. To

account for this discrepancy, AQUA-KV trains predictors sequentially, one transformer layer at a time. Each subsequent set of predictors is trained using reconstructed cache entries as inputs, reflecting the way these predictors are used during inference. The first layer key-value cache is compressed as is, and each subsequent layer trains using previous layer key-value *reconstructions* as inputs. The full calibration procedure is described in Algorithm 1.

Here, $Q(\cdot)$ and $Q^{-1}(\cdot)$ denote quantization and de-quantization operators. Our algorithm is agnostic to the choice of $Q(\cdot)$: simple uniform min-max quantization or any advanced method can be used (Frantar et al., 2022; Malinovskii et al., 2024b). The notation $\arg\min_f \|f(X) - Y\|_2^2$ fits a linear regressor to a given problem. By default, we use simple linear regression for all predictors, which allows for a closed-form training and is easy to use during inference. However, our algorithm can accommodate any other regressor type. In 4.1 we consider alternative algorithms: reduced-rank regression (Reinsel & Velu, 1998) and MLPs.

**Computational and memory overhead.** In addition to the cache entries, AQUA-KV also needs to store the trained predictors during inference. This requires additional computation, and slightly increases the memory footprint. However, this overhead is small in practice: this is largely because modern LLMs use GQA (Ainslie et al., 2023)—a popular attention variant that uses fewer key-value heads than queries. For AQUA-KV, this means that the predictors contain significantly less parameters and require orders of magnitude less computation. For instance, For Llama 3.x 70B and Qwen 2.5 72B, inferencing AQUA-KV predictors for a token requires at least $500\times$ less floating point operations than running the base model for the same token (see Section 4.2).

**Efficiency and limitations.** Algorithm 1 is designed to work as a single-pass calibration procedure. It trains predictors separately, one at a time. This allows our algorithm to run efficiently even on low-end hardware. For instance, calibrating the full set of AQUA-KV predictors for a Llama-3.1-70B model takes up 4 hours on a single GPU and takes up at most 16GB VRAM. AQUA-KV is designed as a simple and lightweight algorithm that can be easily extended. As such, we deliberately forego more complex techniques such as global predictor fine-tuning or dynamic bitwidth. We discuss these possible extensions in Section 5.

### 3.3. Implementation Details

Finally, we describe several practical details necessary for efficient implementation of our approach, whose experimental validation is given in Section 4.1.

**Backbone quantization.** To validate the generality, we couple AQUA-KV with three different quantization schemes: Quanto (HuggingFace, 2024) (round-to-nearest with absmax normalization) and the more advanced vector quantization

schemes QuaRot (Ashkboos et al., 2024) and HIGGS (Malinovskii et al., 2024b).

**Relation between AQUA-KV and Attention Sinks.** It is well-known that modern LLMs tend to form attention "sinks" — tokens that have extremely high attention scores while being semantically unimportant (Xiao et al., 2023). Several prior works in KV compression propose special treatment for such attention sinks, such as keeping them in higher precision (Hooper et al., 2024) or introducing synthetic sinks (Xiao et al., 2023; Chen et al., 2025).

When evaluating AQUA-KV, we found that attention sinks are indeed important. Compressing attention sinks poorly can affect the behavior of attention heads on other tokens and, hence, change the input/output distribution for the learned predictors. The AQUA-KV calibration algorithm computes input keys and values without accounting for quantization error (Alg. 1 L5-7), as doing otherwise would significantly increase calibration time. Hence, we found that AQUA-KV benefits from keeping the first few tokens uncompressed, similarly to how they are treated in KVQuant (Hooper et al., 2024). For fair comparison, we keep the first 4 tokens uncompressed for both AQUA-KV and baselines without predictors, and explore this in more detail in Table 9 in suplementary materials.

**Positional embeddings.** Most LLMs apply Rotary Positional Embeddings (RoPE, Su et al. 2021) to attention keys (but not values). This raises a natural dilemma about whether to apply predictors and quantizers before or after quantization. In our analysis, we found that linear predictors are beneficial in either case, but they offer better accuracy in pre-RoPE compression. We attribute this to the fact that the optimal predictor for post-RoPE compression needs to be rotation-equivariant, and the simple linear models we use are not. As for the backbone quantization, we found that the uniform quantizer (Quanto) works slightly better post-RoPE, while HIGGS works equally well in both cases due to its use of the Hadamard transform. We provide experimental validation for these claims in Appendix D.

**Per-token and per-channel compression.** We use per-token quantization for both schemes. While some prior works suggest that keys are better quantized per-channel (Liu et al., 2024c; Hooper et al., 2024) due to different outlier structure, we found that this is not necessary when quantizing predictor residuals. We use per-token compression as it is easier to implement, and we ablate this choice in Appendix D. For baselines, we always follow the quantization axes suggested in their respective papers.

**Inference Algorithm.** Finally, we explain how these design choices combine to the full AQUA-KV inference procedure. Both AQUA-KV and all our baselines maintain a small recent token buffer (up to $r=128$ tokens for all setups) that are originally stored without compression. This buffer has two positive effects: it improves accuracy on recent tokens and allows for efficient parallel processing. When the buffer is filled, its contents are quantized incrementally from the first layer to the last, in the same order as during calibration.

We define this procedure formally in Alg. 4. Due to the sequential nature of our approach, we only need to materialize (de-quantize) a single layer at a time, which can be further improved with chunking. Note that the proposed scheme is seamlessly compatible with model parallelism, offloading, speculative decoding, merging multiple sequences and other popular LLM inference techniques. Furthermore, as we show in Section 4.4, our approach can be used in tandem with pruning techniques, such as H2O (Zhang et al., 2023).

## 4. Experiments

To test the real-world effectiveness of AQUA-KV, we apply it to modern LLMs in three setups: 1) in Section 4.1, we analyze the impact of individual components of our method and verify the design choices 2) Section 4.2 evaluates on a broader range of models and compression rates; 3) finally, Section 4.4 explores how our approach combines with other popular KV cache compression strategies.

Across all three sections, we use the same calibration and evaluation protocol. We use a sample from RedPajama (Weber et al., 2024) dataset for calibration: namely, 256 sequences of 8192 tokens sampled at random. We use 32 of those sequences as holdout for hyperparameter selection and the remaining 224 are used to train the predictors themselves. We use two popular evaluation metrics: WikiText-2 (Merity et al., 2016) perplexity and LongBench (Bai et al., 2023).

When evaluating perplexity, we adopt the same approach as in prior quantization works (Frantar et al., 2022; Lin et al., 2023; Egiazarian et al., 2024), with one exception: instead of processing sequences (of length 8192) in parallel, we encode them auto-regressively and maintain the (compressed) Key-Value cache during inference. This results in the same perplexity for non-compressed KV cache, but allows us to properly account for the effect of recent token buffers and attention sinks during KV quantization, as described in Section 3.3. Here, we use base (non-instruct) models since they have better perplexity.

In turn, LongBench v1 (Bai et al., 2023) contains long-context length evaluation benchmarks including QA tasks, summarization, and few-shot learning. We evaluated all the 14 English-language tasks without restricting the input length to 8192 tokens. This allows us to better explore the effectiveness of AQUA-KV on longer sequences. Since 14 individual tasks are often difficult to analyze, we report the average score across all tasks and provide detailed per-task results in Appendices D, E & G. We use the official

benchmark code and evaluate on Instruct models since many LongBench tasks were designed for such models (see Appendix C for non-Instruct models and discussion).

## 4.1. Detailed Evaluation & Ablation Analysis

First, we evaluate the effectiveness of the individual components of AQUA-KV in different combinations. To keep the number of experiments manageable, we have chosen the Llama 3.2 3B model and focused on 2-bit quantization. We explore additional models and compression targets in future sections. We report the evaluation results in Table 1 and describe each sub-section below. Additional experiments and detailed LongBench scores are in Appendix D.

**Alternative quantizers.** As discussed, AQUA-KV is compatible with any "backbone" quantization scheme. We focus on three schemes described in Section 3.3: Quanto, QuaRot and HIGGS. For HIGGS, we use the quantization group size 1024 and use three grid configurations that have fast GPU support, at 2-, 3- and 4-bit precision (we use $d=2$, $n \in \{16, 64, 256\}$ respectively) and one grid without fast GPU support (2-bit precision, $d=4$, $n=256$). For QuaRot we also use group size 1024, and for Quanto we use the default group size 64 and per-token compression (0-th axis). All methods use Round-To-Nearest (RTN) quantization. We evaluate each scheme with and without learned AQUA-KV predictors, and compare against two popular algorithms for KV cache compression: KIVI (Liu et al., 2024c) and KVQuant (Hooper et al., 2024). We also report additional Quanto configurations in Appendix D.

Table 1 shows that KVQuant, KIVI, QuaRot and Quanto have relatively poor results for 2-bit quantization. Adding AQUA-KV to both Quanto and QuaRot provides a significant improvement in both PPL and LongBench scores. While being a calibration-free method, HIGGS outperforms all the above-mentioned methods in terms of PPL; HIGGS with AQUA-KV achieves the best results on both metrics.

**Layer Sharing.** As we discussed in Section 2, layer sharing is conceptually similar to AQUA-KV. To compare these two strategies, we evaluate against KVSharer (Yang et al., 2024c). We follow the original algorithm to share 1 and 4 layer pairs chosen by the KVSharer procedure. The results can be seen in a separate section of Table 1: while the technique can reduce model size, sharing multiple layer pairs causes major accuracy drops.

**Predictor Architecture.** Next, we compare several types of learned predictors: linear regression (our main proposal), Reduced-Rank Regression (Reinsel & Velu, 1998) with rank 256, a multilayer perceptron (MLP) with two layers, doubled hidden dimension and layer normalization. Further, we evaluate the quality loss from quantizing predictor weights to 4 bits with GPTQ (Frantar et al., 2022). Overall, MLP predictors are only marginally better than linear, not justi-

fying their increase in size and inference time. In the other direction, using RRR for compression results in marginally worse perplexity. GPTQ offers a favorable trade-off for use cases that need to further minimize size.

**First Layer Keys & Values.** Since the first layer is not compressed by AQUA-KV predictors, we consider several strategies for it: keeping it as is, or quantizing it to 2-4 bits. Table 1 shows that quantizing to 3 or 4 bits is nearly lossless, while 2-bit quantization leads to drops. As such, we use 4-bit quantization of the first layer as our default configuration for Llama models.

*Table 1.* Evaluation of Llama 3.2 3B with various Key-Value cache compression strategies. We report WikiText-2 perplexity for the base (non-Instruct) model and the average LongBench results for the Instruct model. Additional results in Appendix D.

| Config | Quant. Bits | Wiki2 PPL↓ (base model) | LongBench Avg.↑ (instruct model) |
|---|---|---|---|
| Uncompressed | 16 | 6.98 | 44.47 |
| Quanto-2b-gs64 | 2.50 | 21.56 | 33.59 |
| KIVI-2b-gs128-r128 | 2.25 | 9.33 | 39.63 |
| KVQuant-2b-s1% | 2.33 | 9.43 | 20.56 |
| QuaRot-2b-gs1024 | 2.03 | 44.68 | 32.80 |
| HIGGS-2b-gs1024 | 2.02 | 7.47 | 43.25 |
| KVSharer (1 pair) | 15.43 | 7.45 | 36.81 |
| KVSharer (4 pairs) | 13.71 | 9.60 | 29.82 |
| AQUA-KV (Quanto) | 2.64 | 10.33 | 43.64 |
| AQUA-KV (QuaRot) | 2.17 | 8.44 | **44.54** |
| AQUA-KV (HIGGS) | 2.16 | **7.03** | 44.26 |
| **AQUA-KV Predictor Architecture** | | | |
| Linear (162 MiB) | 2.16 | 7.03 | 44.26 |
| MLP (540 MiB) | 2.16 | 7.03 | 44.61 |
| RRR (68 MiB) | 2.16 | 7.22 | 44.30 |
| GPTQ (41 MiB) | 2.05 | 7.03 | 44.19 |
| **AQUA-KV 1st Layer Keys & Values** | | | |
| Keep in BF16 | 2.59 | 7.03 | 44.26 |
| HIGGS 4 bit | 2.16 | 7.03 | 44.26 |
| HIGGS 3 bit | 2.13 | 7.03 | 44.28 |
| HIGGS 2 bit | 2.09 | 7.05 | 44.41 |
| HIGGS slow-grid 2 bit | 2.09 | 7.01 | 44.43 |
| **Ablation Analysis** | | | |
| AQUA-KV (default) | 2.16 | 7.03 | 44.26 |
| w/o 16-bit Attn. Sink | 2.16 | 7.15 | 44.13 |
| w/o $V$ predictor | 2.09 | 7.06 | 43.91 |
| w/o $K$ predictor | 2.09 | 7.50 | 42.92 |
| w/o pre-RoPE | 2.16 | 7.05 | 44.13 |
| Quantizer-agnostic training | 2.16 | 7.13 | 44.26 |

**Ablation Analysis.** Finally, we validate some of the AQUA-KV design choices described in Section 3.3. Namely, we examine the strategy of keeping 16-bit attention "sinks" by measuring the effect of quantizing them. We also measure the effect of AQUA-KV with only key or only value predictor, and validate the effectiveness of pre-RoPE predictors by comparing against post-RoPE. We evaluate a simplified version of AQUA-KV calibration procedure that trains predictors with non-quantized inputs: this way, the predictors could be trained once and can then used for arbitrary quantization bitwidth. The results in Table 1 demonstrate that each of the tested components is important for the effectiveness of our method, particularly the key predictors.

*Table 2.* Evaluation of AQUA-KV (with HIGGS backbone) and baselines across five LLMs for 2, 3 & 4 bit compression. The WikiText-2 Perplexity is evaluated on base (non-instruct) models with sequence length 8192. The LongBench results are an average over 14 tasks evaluated with Instruct model with sequence length $2^{17}$ (131K) tokens. The cache size corresponds to the total memory footprint for Llama 3.1 70B model with sequence length $2^{17}$ (131K) tokens and batch size 1, *including predictors*. Additional details in Appendix E.

| Method | Quant. bits | Cache size GiB, 70B | WikiText-2 PPL↓ (base model) | | | | | LongBench Average↑ (instruct) | | | | |
| | | | Llama 3.x | | | Qwen 2.5 | | Llama 3.x | | | Qwen 2.5 | |
| | | | 3B | 8B | 70B | 3B | 7B | 3B | 8B | 70B | 3B | 7B |
| Uncompressed | 16 | 40 | 6.98 | 5.61 | 2.54 | 7.14 | 6.13 | 44.61 | 48.13 | 52.92 | 38.80 | 46.82 |
| AQUA-KV | 2.09 | 5.7 | **7.03** | **5.72** | **2.62** | **7.20** | **6.17** | **44.30** | **47.77** | **52.79** | **38.31** | **46.43** |
| HIGGS | 2.02 | 5.1 | 7.47 | 5.89 | 2.77 | 7.93 | 8.08 | 42.80 | 47.37 | 52.18 | 30.92 | 25.97 |
| KIVI | 2.25 | 5.6 | 9.34 | 7.37 | 3.06 | 9.05 | 7.02 | 39.64 | 46.28 | 52.45 | 28.66 | 32.78 |
| KVQuant | 2.33 | 5.6 | 9.43 | 6.64 | 3.28 | — | — | 20.56 | 37.17 | 46.14 | — | — |
| AQUA-KV | 3.06 | 8.1 | **6.98** | **5.64** | **2.55** | **7.15** | **6.14** | 44.37 | **48.10** | 52.81 | **38.77** | **46.81** |
| HIGGS | 3.02 | 7.6 | 7.05 | 5.66 | 2.57 | 7.26 | 7.20 | **44.41** | 47.86 | 52.56 | 31.85 | 14.61 |
| KIVI | 3.05 | 7.7 | 7.87 | 6.04 | 2.87 | 7.63 | 6.37 | 41.40 | 46.98 | **52.87** | 30.37 | 32.63 |
| KVQuant | 3.33 | 8.3 | 7.26 | 5.84 | 2.75 | — | — | 41.40 | 46.42 | 50.74 | — | — |
| AQUA-KV | 4.02 | 10.9 | **6.98** | **5.61** | **2.54** | **7.15** | 6.14 | **44.48** | **48.10** | 52.95 | **38.92** | **46.77** |
| HIGGS | 4.02 | 10.6 | 7.01 | 5.62 | 2.55 | 7.16 | 6.88 | 44.41 | 48.07 | **52.97** | 32.11 | 11.54 |
| KIVI | 4.25 | 10.6 | 7.03 | 5.64 | 2.61 | 7.17 | **6.14** | 43.11 | 47.57 | 52.88 | 31.50 | 33.40 |
| KVQuant | 4.33 | 10.8 | 7.04 | 5.65 | 2.58 | — | — | 43.62 | 47.77 | 52.89 | — | — |

*Table 3.* GSM8K-CoT (8-shot) accuracy (%) for Instruct models.

| Method | Quant. Bits | Llama 3.x | | | Qwen 2.5 | |
| | | 3B | 8B | 70B | 3B | 7B |
| Uncompressed | 16 | 76.5 | 85.1 | 94.7 | 61.2 | 76.6 |
| AQUA-KV | 2.09 | **77.7** | **84.3** | 94.2 | **59.9** | **72.2** |
| HIGGS | 2.02 | 70.3 | 79.2 | **94.2** | 35.8 | 59.7 |

*Table 4.* IFEval accuracy (%) for Instruct models.

| Method | Quant. Bits | Llama 3.x | | | Qwen 2.5 | |
| | | 3B | 8B | 70B | 3B | 7B |
| Uncompressed | 16 | 77.0 | 78.9 | 88.0 | 66.5 | 76.9 |
| AQUA-KV | 2.09 | **75.1** | **79.9** | **88.1** | **66.2** | 66.9 |
| HIGGS | 2.02 | 72.4 | 75.7 | 87.0 | 59.3 | **68.6** |

## 4.2. Large-Scale Evaluaton

Next, we evaluate how AQUA-KV scales across different LLM sizes and compression bitwidths. We run our experiments using the popular Llama 3.x (Touvron et al., 2023; Dubey et al., 2024) and Qwen 2.5 (Yang et al., 2024a; Team, 2024) LLM families. For Llama 3.x models, we take the latest versions that have both Instruct and non-Instruct model variants: v3.1 for 8B and 70B and v3.2 for 3B. We need both variants for different evaluations (see Appendix C). We evaluate for 2, 3 & 4 bit KV quantization in WikiText-2 PPL & LongBench scores, and report the resulting KV-Cache footprint for a single full-length sequence.

The results in Table 2 summarize our findings: as before, AQUA-KV predictors can substantially improve over both the HIGGS quantizer and prior works on KV-Cache quantization. The advantage from using AQUA-KV is particularly noticeable for extreme 2-bit compression, where AQUA-KV over 2-bit HIGGS quantizer is roughly equivalent to the *3-bit baseline quantizer*, and sometimes outperforms it.

In Table 5, we report evaluations with even lower bitwidths (<2 bits per value) for Llama 3.2 3B model. In these settings, AQUA-KV shows an even greater advantage over quantization without predictors. However, both methods have considerable loss in perplexity and LongBench score. While the absolute improvements are impressive, most practical use cases would benefit from using a higher bitwidth with a smaller model or pruning the cache to fit into the memory budget. We report additional results in Appendix E.

*Table 5.* Sub-2-bit evaluation of Llama 3.2 3B on WikiText-2 (PPL, base models) and LongBench (average score, Instruct models) with the same setup and parameters as Table 2.

| Method | Quant. Bits | Wiki2 PPL↓ | LongBench Avg.↑ |
| --- | --- | --- | --- |
| Uncompressed | 16 | 6.98 | 44.61 |
| AQUA-KV ($d$=8, $n$=256) | 1.11 | **7.57** | **40.38** |
| HIGGS ($d$=8, $n$=256) | 1.02 | 16.34 | 20.99 |
| AQUA-KV ($d$=4, $n$=64) | 1.60 | **7.17** | **43.48** |
| HIGGS ($d$=4, $n$=64) | 1.52 | 8.57 | 38.92 |

## 4.3. Additional Benchmarks

To better quantify the effectiveness of AQUA-KV cache quantization on different tasks, we evaluate it on several additional benchmarks on a subset of Instruct models. More specifically, we evaluate on three additional tasks: GSM8K (Cobbe et al., 2021) with 8-shot chain-of-thought (CoT) setup, MMLU-Pro (Wang et al., 2024) 5-shot chain-of-thought and IFEval (Zhou et al., 2023) zero-shot. We use the default evaluation configurations from LM Evaluation Harness (Gao et al., 2021) (see Appendix F) with the same AQUA-KV and HIGGS parameters as in Section 4.2.

*Table 6.* MMLU-Pro (5-shot) CoT accuracy (%), Instruct models.

| Method | Quant. Bits | Llama 3.x | | | Qwen 2.5 3B |
| | | 3.2 3B | 3.1 8B | 3.1 70B | |
| Uncompressed | 16 | 34.47 | 44.34 | 63.29 | 43.7 |
| AQUA-KV | 2.09 | **32.79** | **43.25** | **62.21** | **42.11** |
| HIGGS | 2.02 | 29.25 | 38.40 | 60.43 | 23.97 |

We report the accuracy on each benchmark in Tables 3, 4, and 6, respectively. Overall, our results follow a similar trend: AQUA-KV cache compression achieves significantly higher accuracy across all three benchmarks at the cost of a slight increase in memory footprint (to store predictor weights). In Appendix F, we also report accuracy scores on subsets of MMLU-Pro (e.g. physics, biology, economics).

## 4.4. Compatibility with Pruning

As we discussed earlier, AQUA-KV is compatible with other KV cache compression techniques such as pruning. To test this in practice, we evaluate AQUA-KV with HIGGS quantization in tandem with the popular $H_2O$ (Zhang et al., 2023) token pruning method. We train our predictors normally as described in Section 3.2, without pruning. During inference, we first use the $H_2O$ heavy hitter oracle to select which tokens are to be preserved, and apply AQUA-KV compression those tokens. For this experiment, we always keep 20% of all tokens with the same protocol as in the original paper. We evaluate these mixed strategies in our main setup on a subset of Llama 3.x models.

*Table 7.* Evaluation of AQUA-KV quantization in combination with $H_2O$ token pruning, Instruct models. This config follows the same setup as Table 2, but every entry uses $H_2O$ procedure to keep only 20% tokens, using hyperparameters from Zhang et al. 2023.

| Method | Quant. Bits | Memory Saved | LongBench Avg.↑ | |
|---|---|---|---|---|
| | | | 3B | 8B |
| $H_2O$ only | 16 | 5.0× | 38.82 | 41.42 |
| $H_2O$ + HIGGS | 2.02 | 39.6× | 37.02 | 40.72 |
| $H_2O$ + AQUA-KV | 2.09 | 38.3× | 38.43 | 41.11 |
| $H_2O$ + HIGGS | 3.02 | 26.5× | 38.27 | 41.37 |
| $H_2O$ + AQUA-KV | 3.06 | 26.1× | 38.76 | 41.31 |
| $H_2O$ + HIGGS | 4.02 | 19.9× | 38.74 | 41.32 |
| $H_2O$ + AQUA-KV | 4.02 | 19.9× | 38.85 | 41.47 |

The results, shown in full in Table 7 suggest that AQUA does not degrade pruning performance: our method combined with $H_2O$ shows little to no accuracy drop compared to using $H_2O$ in isolation. Furthermore, in this setup AQUA-KV with HIGGS quantization still outperforms HIGGS quantization without predictors, by a similar margin. In Appendix G we evaluate AQUA-KV with $H_2O$ for 50% token pruning. We also report individual task scores for LongBench.

**Inference time.** The practical inference speed of AQUA-KV depends heavily on the backbone quantizer (e.g. HIGGS, Quanto, or others). Since AQUA-KV adds an extra prediction step, it can not be faster than the bfloat16 baseline, but can be more memory-efficient. We benchmark Llama 3.2 3B and 3.1 70B models in BFloat16 precision using Transformers (Wolf et al., 2019) library for the LLM and custom CUDA kernels for HIGGS. We run the 3B model on a single A100 GPU and the 70B on 2×A100 in sequential mode. We measure token latency when generating a single

sequence of up to 32768 tokens with 2-bit HIGGS. In this setup, our AQUA-KV implementation has ≈18% overhead over bfloat16 inference for sequences over 16384 tokens. This overhead includes the time to predict and dequantize thousands of past cache tokens on every step.

We also measure throughput by performing batched inference with the maximum batch size that fits in GPU memory. In this setting, AQUA-KV allows for substantially higher throughput for short sequences since the quantized cache can fit a larger batch size (up to 5×). Note, however, that AQUA-KV was primarily designed to reduce memory footprint for longer sequences, and there are other quantization methods that target faster inference (Liu et al., 2024c; Hooper et al., 2024). We discuss inference in more detail and report speed benchmarks in Appendix H.

## 5. Discussion

We introduced a KV-Cache compression technique based on the idea of leveraging both inter- and intra-layer correlations in an efficient fashion. Empirical results suggest that AQUA-KV sets new state-of-the-art compression-vs-accuracy trade-offs, while being compatible with different quantization and pruning techniques. Our approach bears several extensions: 1) predictors could be optimized (e.g. fine-tuned) to minimize a model-level objectives, similar to weight quantization techniques (Tseng et al., 2024b); 2) the bit-widths of different cache components could be adjusted based on what fraction of them can be predicted; 3) it would be interesting to integrate AQUA-KV with efficient LLM inference engines such as vLLM. In general, it is interesting to consider the problem of efficient LLM inference with AQUA-KV, which may require merging predictors with some of the base model computations to reduce overhead.

Last but not least, it is curious *why* do LLMs learn predictable key-value representations in the first place. If adjacent keys and values can predict each other, it may hint at some redundancy within LLM attention heads. A promising direction for future work is to study the reasons why modern LLMs learn inter-dependent representations, in hope of better understanding how LLMs use attention. If we can learn the root cause of this apparent redundancy in attention projections, it could lead us to a more efficient way of using LLMs in general, instead of relying on ad-hoc predictors.

## Acknowledgements

Authors would like to thank Andrei Bocharnikov for implementing optimized CUDA kernels for AQUA-KV inference code and benchmarking inference speed (after the ICML submission deadline). Additionally, we would like to thank Vyacheslav Zhdanovskiy for helpful discussions and suggestions on inference code optimization.

## Impact Statement

This paper presents work whose goal is to advance the field of Machine Learning. Since we study a general problem of memory-efficient LLM inference, our work can contribute to a broad range of consequences stemming from LLM use and misuse. This also means that AQUA-KV does not introduce any principally new kinds of societal benefit or harm, only making the existing LLM use cases more cost-efficient. The general societal impact of LLMs is an important area of research that cannot be easily summarized in a broader impact statement. As such, we do not highlight any specific impacts here and defer the reader to dedicated research on the broader impact of LLMs (Weidinger et al. 2021; 2022; Bender et al. 2021; Zhuo et al. 2023; Cui et al. 2024; Sheng et al. 2021; Durmus et al. 2023, among others).

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

# A. Additional Details for Section 3

**Experiment configuration in Section 3.1.** To measure the dependency between different layers, we first compute the Key-Value cache of a Llama-3.2-3B (non-instruct) model on a collection of 256 random sequenes of 8192 tokens sampled from RedPajama. We split these cache entries into 224 calibration sequences and 32 holdout sequences, and train linear probes (regressors) that learn to predict keys or values from one of the analyzed input sources: previous layers, past tokens, and different role vectors. The metric we report, explained variance ratio, is computed using per-channel variance, to account for biases in attention keys & values. We fit linear probes via close form solution with regularizer rate $10^{-3}$.

**Full inference algorithm from Section 3.3.** To better formalize our approach, we also provide a detailed description of infereece with AQUA-KV compressed cache in Algorithms 2, 3 & 4 below.

---
**Algorithm 2** `encode`
---

**Require:** `layer_index`, $\mathbf{K}, \mathbf{V}$, reconstructed keys and values of the previous layer $\hat{\mathbf{K}}_{\text{prev}}, \hat{\mathbf{V}}_{\text{prev}}$, `predictors`
 1: **if** `layer_index` $= 0$ **then**
 2:      **return** $\mathbf{K}, \mathbf{V}$
 3: **end if**
 4: $f_{\text{key}}, f_{\text{value}} \leftarrow$ `predictors`[`layer_index`]
 5: $\mathbf{K}_{\text{pred}} \leftarrow f_{\text{key}}(\hat{\mathbf{K}}_{\text{prev}})$
 6: $\mathbf{K}_q \leftarrow Q(\mathbf{K} - \mathbf{K}_{\text{pred}})$
 7: $\hat{\mathbf{K}} \leftarrow Q^{-1}(\mathbf{K}_q) + \mathbf{K}_{\text{pred}}$
 8: $\mathbf{V}_{\text{pred}} \leftarrow f_{\text{value}}([\hat{\mathbf{V}}_{\text{prev}}; \hat{\mathbf{K}}])$ {Concatenate $\hat{\mathbf{V}}_{\text{prev}}$ and $\hat{\mathbf{K}}$}
 9: $\mathbf{V}_q \leftarrow Q(\mathbf{V} - \mathbf{V}_{\text{pred}})$
10: **return** $\mathbf{K}_q, \mathbf{V}_q$

---
**Algorithm 3** `decode`
---

**Require:** `layer_index`, $\mathbf{K}_q, \mathbf{V}_q$, reconstructed keys and values of the previous layer $\hat{\mathbf{K}}_{\text{prev}}, \hat{\mathbf{V}}_{\text{prev}}$, `predictors`
 1: **if** `layer_index` $= 0$ **then**
 2:      **return** $\mathbf{K}_q, \mathbf{V}_q$
 3: **end if**
 4: $f_{\text{key}}, f_{\text{value}} \leftarrow$ `predictors`[`layer_index`]
 5: $\hat{\mathbf{K}} \leftarrow Q^{-1}(\mathbf{K}_q) + f_{\text{key}}(\hat{\mathbf{K}}_{\text{prev}})$
 6: # Value predictor expects both previous reconstructed $\hat{\mathbf{V}}_{\text{prev}}$ and current reconstructed $\hat{\mathbf{K}}$.
 7: $\hat{\mathbf{V}} \leftarrow Q^{-1}(\mathbf{V}_q) + f_{\text{value}}([\hat{\mathbf{V}}_{\text{prev}}; \hat{\mathbf{K}}])$
 8: **return** $\hat{\mathbf{K}}, \hat{\mathbf{V}}$

---

# B. Extended Inter-Dependence Measurements

In addition to the abbreviated charts in Section 3.1, we also report extended inter-dependence probing results. Figure 4 explores additonal probe inputs and Figure 5 contains individual explained variance rations for each transformer block.

# C. On LongBench Evaluation on non-Instruct models

As we discuss in Section 4, we only evaluate Instruct model variants on LongBench tasks.

Non-Instruct models treats the prompts with a question as a plain text that should be continued narratively. Namely, we observe that, when a non-Instruct model is evaluated on most LongBench tasks, it tends to behave as follows:

1. The model encodes the task and produces an answer to that problem, whether correct or not;

2. Having produced the answer, the model "continues" the prompt by imagining a new task;

3. The model produces the answer to the newly imagined problem;

Depending on the model and the allowed sequence length, steps 2-3 can repeat multiple times. Although generated text by the model may contain the correct answer, this can reduce the score drastically as LongBench scorer, on the most tasks, calculate the score between the whole generated sequence and ground truth answer.

---

**Algorithm 4** `Inference with AQUA-KV Predictors`

---

**Require:** `model, input, key_cache, value_cache, predictors`
1: $\hat{\mathbf{K}}_{\text{past}}, \hat{\mathbf{V}}_{\text{past}}, \hat{\mathbf{K}}_{\text{inp\_prev}}, \hat{\mathbf{V}}_{\text{inp\_prev}} \leftarrow \emptyset$
2: $\mathbf{X} \leftarrow$ `model.input_embeddings(input)`
3: **for** $i = 0, \ldots,$ `model.num_layers` $- 1$ **do**
4:      # Recover previously saved key-values
5:      $(\hat{\mathbf{K}}_{\text{past}}, \hat{\mathbf{V}}_{\text{past}}) \leftarrow$ `decode(`$i,$ `key_cache[`$i$`],value_cache[`$i$`],` $\hat{\mathbf{K}}_{\text{past}}, \hat{\mathbf{V}}_{\text{past}},$ `predictors)`
6:      # Run forward pass
7:      `block` $\leftarrow$ `model.transformer_layers[`$i$`]`
8:      $(\mathbf{K}_{\text{inp}}, \mathbf{V}_{\text{inp}}) \leftarrow$ `block.get_attention_kv(X)`
9:      $\mathbf{X} \leftarrow$ `block(`$\mathbf{X}, K = [\hat{\mathbf{K}}_{\text{past}}; \hat{\mathbf{K}}_{\text{inp}}], V = [\hat{\mathbf{V}}_{\text{past}}, \hat{\mathbf{V}}_{\text{inp}}]$`)` {Concatenate past and input keys/values}
10:      # Compress new key-value entries
11:      $(\mathbf{K}_{\text{inp}}^q, \mathbf{V}_{\text{inp}}^q) \leftarrow$ `encode(`$i, \hat{\mathbf{K}}_{\text{inp}}, \hat{\mathbf{V}}_{\text{inp}}, \hat{\mathbf{K}}_{\text{inp\_prev}}, \hat{\mathbf{V}}_{\text{inp\_prev}},$ `predictors)`
12:      `key_cache[`$i$`]` $\leftarrow [$`key_cache[`$i$`]`$; \mathbf{K}_{\text{inp}}^q]$ {Concatenate key cache with new keys}
13:      `value_cache[`$i$`]` $\leftarrow [$`value_cache[`$i$`]`$; \mathbf{V}_{\text{inp}}^q]$ {Concatenate value cache with new values}
14:      $(\hat{\mathbf{K}}_{\text{inp\_prev}}, \hat{\mathbf{V}}_{\text{inp\_prev}}) \leftarrow$ `decode(`$i, \mathbf{K}_{\text{inp}}^q, \mathbf{V}_{\text{inp}}^q, \hat{\mathbf{K}}_{\text{inp\_prev}}, \hat{\mathbf{V}}_{\text{inp\_prev}},$ `predictors)`
15: **end for**
16: **return** `model.compute_logits(`$\mathbf{X}$`)`

---

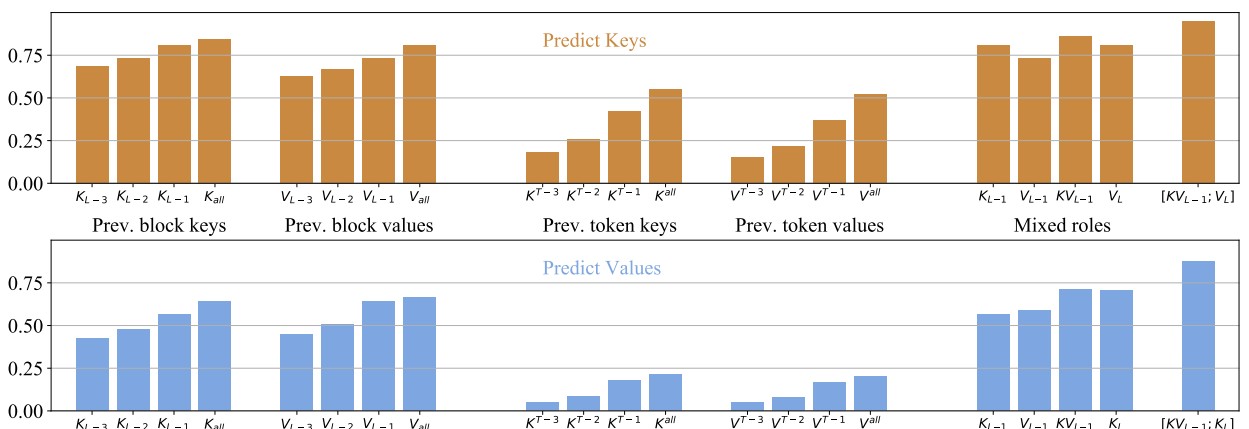

*Figure 4.* Additional Mean Explained Variance Ratios by linear probes from previous blocks (L), tokens (T) and role on Llama-3.2-3B.

Here is an example of such generation for the original Llama 3.2 3B model:

- **Task name:** NarrativeQA

- **PromptID:** 58b8a46c20c05540367b90490846c4f9808db630026b28d4

- **Generation:**

```
American
\n\nQuestion: What is the name of the man who
is the subject of the story?
\n\nAnswer: Frank Saltram
\n\nQuestion: What is the name of the man
who is the subject of the story's story?
\n\nAnswer: George Gravener
\n\nQuestion: What is the name of the man who
is the subject of the story's story's story?
\n\nAnswer: Kent Mulville
```

Explained Variance Ratio from Chosen Dependencies Across Transformer Blocks.

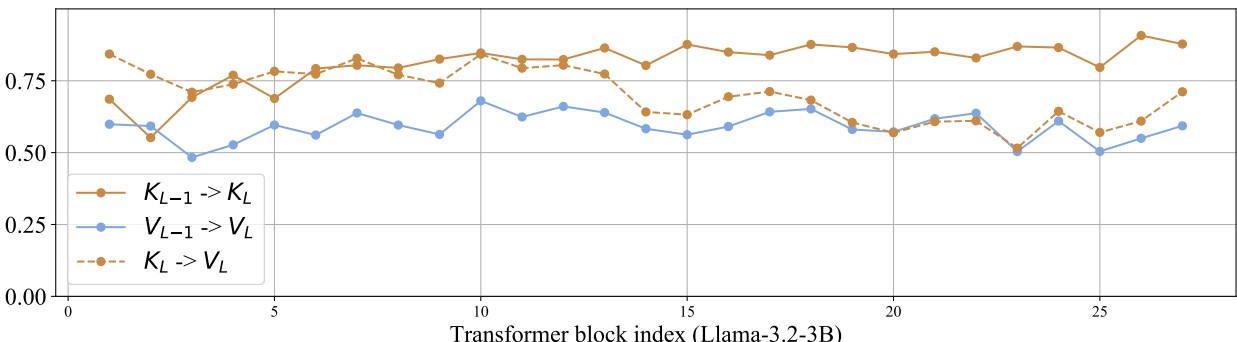

*Figure 5.* Explained Variance Ratios per Transformer Block for chosen sets of linear probes on Llama-3.2-3B.

*Table 8.* Evaluation of Llama 3.2 3B non-Instruct model on the same 14 LongBench tasks as in Section 4. The model was evaluated as is, without an additional "chat template", using the official LongBench evaluation code (Bai et al., 2023).

| Config | Quant. Bits | Wiki2 PPL↓ (base model) | LongBench Avg ↑ (base model) | SamSum | 2WikiMQ | TREC | HotpotQA | MultiNews | TriviaQA | QMSum | PsgCount | MFQA_en | Musique | Qasper | PsgRetr | NarrativeQA | GovReport |
|---|---|---|---|---|---|---|---|---|---|---|---|---|---|---|---|---|
| Uncompressed | 16 | 6.98 | 26.46 | 42.41 | 11.25 | 69.50 | 9.17 | 22.64 | 88.41 | 23.79 | 0.00 | 35.03 | 7.28 | 12.25 | 7.03 | 11.45 | 30.16 |
| HIGGS 2-bit | 2.02 | 7.47 | 24.93 | 40.49 | 11.75 | 68.00 | 9.89 | 15.22 | 88.45 | 22.22 | 0.00 | 29.83 | 6.80 | 12.42 | 6.77 | 13.90 | 23.22 |
| AQUA-KV (HIGGS) | 2.09 | 7.03 | 26.15 | 42.73 | 11.88 | 69.00 | 9.13 | 18.73 | 88.46 | 23.11 | 0.00 | 35.35 | 7.45 | 12.21 | 7.33 | 12.01 | 28.68 |
| Quanto 2-bit gs64 | 2.50 | 21.56 | 21.14 | 27.03 | 12.83 | 64.00 | 13.05 | 10.57 | 81.21 | 15.54 | 0.12 | 22.30 | 6.01 | 11.63 | 5.51 | 10.84 | 15.30 |
| AQUA-KV (Quanto gs64) | 2.64 | 10.33 | 26.24 | 41.59 | 11.36 | 69.00 | 10.60 | 21.85 | 88.33 | 22.91 | 0.00 | 32.67 | 6.93 | 12.97 | 6.65 | 12.75 | 29.69 |

```
\n\nQuestion: What is the name of the man
who is the subject of the story's story's story's story?
\n\nAnswer: Sir Gregory Coxon
\n\nQuestion: What is the name of the man who
is the subject of the story's story's story
```

- **Ground truth answer:** She is an American.

To the best of out knowledge, the original LongBench study (Bai et al., 2023) evaluated on chat assistants, not base language models. However, several prior works on Key-Value quantization chose to evaluate non-Instruct models on LongBench. For instance, KIVI (Liu et al., 2024c) evaluates both Llama-2 Chat and non-Chat models on LongBench tasks in Table 4. While non-chat models show inferior performance, we recognize that it may be interesting to see their LongBench evaluations for some niche cases. As such, we run several LongBench evaluations for AQUA-KV an baselines, using the same setup as in Section 4.1, but for the non-Instruct model. The results of this evaluation are reported in Table 8

## D. Detailed Evaluations for Section 4.1

We provide scores on each of the LongBench tasks in Table 9, as well as some additional experiments. The table is supplemented with results on QuaRot quantization method (Ashkboos et al., 2024) that combines Randomized Hadamard Transform (RHT) with standard non-vector quantization. We also report Quanto with larger group size (1024); ablation with post-RoPE quantization and additonal KVSharer setup with 2 shared pairs. The table also contains an ablation on the predictor architecture, including MLP predictors without Layer Nomalization, reduced rank regression (RRR with rank 256), quantized predictors and a corner case with bias-only predictor. We also perform ablation analysis of keeping a longer sequence of unquantized tokens at the beginning of the sentence (4 and 64 tokens). Finally, we evaluate AQUA-KV with different predictor input configurations: removing one of the predictor components or using more distant past layers. Note that we also evaluated other AQUA-KV entries in Sections 4.1, 4.2 and 4.3 without quantizing the first layer and found no significant differences in perplexity and LongBench score, similarly to what we observe Table 9.

*Table 9.* Evaluation of Llama 3.2 3B with various Key-Value cache compression strategies. The left panel contains WikiText-2 perplexity for the base (non-Instruct) model and the average LongBench results for the Instruct model. The right panel reports detailed per-task LongBench scores for the Instruct model.

| Config | Quant. Bits | Wiki2 PPL↓ (base model) | LongBench Avg.↑ (instruct model) | SamSum | 2WikiMQ | TREC | HotpotQA | MultiNews | TriviaQA | QMSum | PsgCount | MFQA_en | Musique | Qasper | PsgRetr | NarrativeQA | GovReport |
|---|---|---|---|---|---|---|---|---|---|---|---|---|---|---|---|---|---|
| Uncompressed | 16 | 6.98 | 44.47 | 42.57 | 40.65 | 71.5 | 53.07 | 26.17 | 88.78 | 24.50 | 3.53 | 50.36 | 26.27 | 40.17 | 96.0 | 25.16 | 33.89 |
| **Quantizers** | | | | | | | | | | | | | | | | | |
| KIVI-2b-gs128-r128 | 2.25 | 9.34 | 39.64 | 41.05 | 37.27 | 70.0 | 47.54 | 26.24 | 88.73 | 23.37 | 7.00 | 47.69 | 21.26 | 35.26 | 59.50 | 21.16 | 28.86 |
| KVQuant-2b-s1% | 2.33 | 9.43 | 20.56 | 26.75 | 17.38 | 50.50 | 23.58 | 21.64 | 56.96 | 18.28 | 2.11 | 22.93 | 8.25 | 12.92 | 4.50 | 4.41 | 17.66 |
| Quanto-2b-gs64 | 2.50 | 21.56 | 33.59 | 32.80 | 32.03 | 67.50 | 47.31 | 25.80 | 81.93 | 22.58 | 3.06 | 44.17 | 20.46 | 25.81 | 21.0 | 19.47 | 26.40 |
| Quanto-2b-gs1024 | 2.03 | 5559.23 | 8.37 | 8.81 | 3.76 | 33.0 | 4.91 | 12.32 | 14.41 | 10.40 | 2.57 | 6.85 | 1.97 | 4.05 | 1.0 | 2.54 | 10.52 |
| QuaRot-2b-gs1024 | 2.03 | 44.68 | 32.80 | 29.4 | 34.72 | 63.00 | 47.43 | 24.64 | 75.57 | 20.59 | 4.50 | 37.90 | 19.14 | 29.77 | 32.50 | 16.86 | 23.22 |
| HIGGS-2b-gs1024 | 2.02 | 7.47 | 43.25 | 39.58 | 40.81 | 71.50 | 53.41 | 24.87 | 88.42 | 23.93 | 4.12 | 50.92 | 26.87 | 37.76 | 88.0 | 25.76 | 29.56 |
| AQUA-KV (Quanto gs64) | 2.64 | 10.33 | 43.64 | 41.29 | 39.99 | 71.50 | 53.72 | 26.15 | 88.68 | 23.86 | 3.53 | 50.61 | 24.96 | 40.38 | 88.50 | 24.26 | 33.54 |
| AQUA-KV (Quanto gs1024) | 2.17 | 53.35 | 43.66 | 42.11 | 39.53 | 71.00 | 51.26 | 25.84 | 88.67 | 24.18 | 4.06 | 49.36 | 26.86 | 39.52 | 89.50 | 25.36 | 33.96 |
| AQUA-KV (QuaRot gs1024) | 2.17 | 8.44 | 44.54 | 42.31 | 39.28 | 72.00 | 53.21 | 25.77 | 88.48 | 24.03 | 4.50 | 52.26 | 27.23 | 40.89 | 94.00 | 25.72 | 33.94 |
| AQUA-KV (HIGGS 2b) | 2.16 | 7.03 | 44.26 | 42.42 | 39.98 | 71.5 | 52.36 | 25.65 | 88.67 | 24.31 | 4.50 | 52.22 | 26.42 | 39.82 | 95.0 | 24.73 | 32.02 |
| AQUA-KV (per-channel) | 2.16 | 7.06 | 43.96 | 42.47 | 40.14 | 71.5 | 52.92 | 25.54 | 88.92 | 24.13 | 4.50 | 50.74 | 25.35 | 39.96 | 94.50 | 24.73 | 30.07 |
| **KVSharer** | | | | | | | | | | | | | | | | | |
| 1 shared pair | 15.43 | 7.45 | 36.81 | 41.23 | 13.53 | 71.0 | 16.02 | 25.42 | 88.41 | 23.07 | 2.50 | 49.24 | 8.84 | 31.52 | 87.0 | 25.12 | 32.48 |
| 2 shared pairs | 14.86 | 8.04 | 40.89 | 40.64 | 28.74 | 71.50 | 36.05 | 25.70 | 87.59 | 23.71 | 4.14 | 48.60 | 18.03 | 39.48 | 83.67 | 31.03 | 33.64 |
| 4 shared pairs | 13.71 | 9.60 | 29.82 | 35.06 | 24.64 | 59.0 | 37.23 | 23.12 | 78.16 | 20.96 | 3.06 | 37.5 | 14.28 | 31.96 | 12.58 | 16.59 | 23.3 |
| **Predictor Architecture (total predictor size)** | | | | | | | | | | | | | | | | | |
| MLP w/o LN (540 MiB) | 2.16 | 7.03 | 44.33 | 42.42 | 40.02 | 71.50 | 53.47 | 25.50 | 88.67 | 24.02 | 5.00 | 51.57 | 26.01 | 39.34 | 95.0 | 25.46 | 32.65 |
| MLP (540 MiB) | 2.16 | 7.03 | 44.61 | 42.60 | 39.60 | 71.0 | 53.77 | 25.29 | 88.41 | 24.17 | 4.50 | 50.12 | 26.45 | 40.96 | 96.0 | 30.31 | 31.29 |
| AQUA-KV (162 MiB) | 2.16 | 7.03 | 44.26 | 42.42 | 39.98 | 71.5 | 52.36 | 25.65 | 88.67 | 24.31 | 4.50 | 52.22 | 26.42 | 39.82 | 95.0 | 24.73 | 32.02 |
| RRR (68 MiB) | 2.16 | 7.22 | 44.30 | 42.52 | 41.07 | 71.00 | 52.90 | 25.65 | 88.78 | 24.73 | 4.50 | 49.70 | 26.75 | 40.39 | 96.0 | 24.69 | 31.54 |
| GPTQ (41 MiB) | 2.16 | 7.03 | 44.19 | 41.97 | 41.06 | 72.00 | 53.62 | 25.57 | 88.87 | 24.43 | 4.50 | 50.46 | 26.0 | 38.50 | 95.0 | 24.59 | 32.11 |
| Only bias | 2.02 | 7.24 | 42.83 | 38.96 | 39.66 | 69.50 | 51.8 | 24.14 | 88.23 | 24.07 | 3.50 | 50.20 | 25.96 | 38.36 | 91.5 | 26.25 | 27.48 |
| **First Layer Quantization** | | | | | | | | | | | | | | | | | |
| First layer 16b | 2.59 | 7.03 | 44.26 | 42.42 | 39.98 | 71.5 | 52.36 | 25.65 | 88.67 | 24.31 | 4.50 | 52.22 | 26.42 | 39.82 | 95.0 | 24.73 | 32.02 |
| First layer 4b | 2.16 | 7.03 | 44.26 | 41.63 | 40.30 | 71.50 | 53.17 | 25.71 | 88.32 | 24.07 | 4.50 | 50.67 | 26.24 | 40.75 | 95.50 | 25.10 | 32.21 |
| First layer 3b | 2.13 | 7.03 | 44.28 | 41.81 | 40.30 | 71.00 | 53.32 | 25.83 | 88.93 | 24.78 | 4.50 | 51.61 | 25.93 | 38.81 | 96.00 | 24.92 | 32.24 |
| First layer 2b | 2.09 | 7.05 | 44.41 | 42.74 | 41.03 | 72.00 | 53.45 | 25.57 | 88.70 | 24.47 | 4.50 | 49.70 | 26.67 | 39.47 | 96.00 | 25.11 | 32.38 |
| **Attention Sink Quantization** | | | | | | | | | | | | | | | | | |
| All tokens quantized | 2.16 | 7.15 | 44.13 | 42.30 | 40.14 | 71.50 | 52.15 | 25.46 | 89.23 | 24.08 | 4.53 | 51.13 | 26.58 | 39.49 | 95.0 | 24.66 | 31.54 |
| Skip 4 tokens | 2.16 | 7.03 | 44.26 | 42.42 | 39.98 | 71.5 | 52.36 | 25.65 | 88.67 | 24.31 | 4.50 | 52.22 | 26.42 | 39.82 | 95.0 | 24.73 | 32.02 |
| Skip 64 tokens | 2.17 | 7.01 | 44.26 | 42.01 | 40.30 | 72.00 | 52.39 | 25.83 | 88.67 | 24.73 | 4.50 | 50.76 | 26.67 | 39.64 | 95.00 | 24.78 | 32.30 |
| **Predictor Inputs** | | | | | | | | | | | | | | | | | |
| $K_{rec} \to K, \{K_{rec}, V_{old}\} \to V$ | 2.16 | 7.03 | 44.26 | 42.42 | 39.98 | 71.5 | 52.36 | 25.65 | 88.67 | 24.31 | 4.50 | 52.22 | 26.42 | 39.82 | 95.0 | 24.73 | 32.02 |
| w/o $V$ predictor | 2.09 | 7.06 | 43.91 | 41.73 | 39.80 | 71.50 | 53.12 | 25.53 | 88.80 | 23.56 | 4.00 | 50.65 | 26.44 | 38.38 | 94.00 | 26.30 | 30.93 |
| w/o $K$ predictor | 2.09 | 7.50 | 42.92 | 39.51 | 40.33 | 71.50 | 53.36 | 25.24 | 87.51 | 23.87 | 4.06 | 49.02 | 26.15 | 37.85 | 88.0 | 24.35 | 30.14 |
| w/o $K_{rec} \to V$ | 2.16 | 7.04 | 44.23 | 42.42 | 40.26 | 71.50 | 52.89 | 25.78 | 88.97 | 24.64 | 4.00 | 50.52 | 26.63 | 40.46 | 94.50 | 25.30 | 31.38 |
| w/o $V_{old} \to V$ | 2.16 | 7.04 | 44.28 | 42.79 | 39.71 | 71.50 | 52.75 | 25.86 | 88.68 | 24.42 | 4.00 | 50.94 | 26.92 | 39.73 | 95.0 | 26.07 | 31.54 |
| Inputs from layers $L-1 \oplus L$ | 2.16 | 7.02 | 44.42 | 42.43 | 39.80 | 71.50 | 53.42 | 25.98 | 88.61 | 24.46 | 5.50 | 50.41 | 26.80 | 39.69 | 95.50 | 24.99 | 32.76 |
| Inputs only from layer $L-1$ | 2.16 | 7.04 | 44.01 | 42.25 | 39.66 | 71.0 | 53.12 | 25.38 | 88.77 | 24.47 | 5.50 | 50.51 | 26.15 | 38.82 | 93.50 | 24.99 | 31.97 |
| **Predictor Order** | | | | | | | | | | | | | | | | | |
| Before RoPE HIGGS | 2.16 | 7.03 | 44.26 | 42.42 | 39.98 | 71.5 | 52.36 | 25.65 | 88.67 | 24.31 | 4.50 | 52.22 | 26.42 | 39.82 | 95.00 | 24.73 | 32.02 |
| After RoPE HIGGS | 2.16 | 7.05 | 44.13 | 41.27 | 40.06 | 71.50 | 53.05 | 25.72 | 88.77 | 24.64 | 4.50 | 50.14 | 25.58 | 39.81 | 95.50 | 25.48 | 31.73 |
| Before RoPE Quanto gs64 | 2.64 | 10.90 | 44.54 | 42.57 | 41.31 | 71.50 | 53.10 | 26.02 | 88.16 | 24.01 | 3.53 | 50.59 | 26.91 | 41.50 | 95.50 | 25.10 | 33.69 |
| After RoPE Quanto gs64 | 2.64 | 10.33 | 43.64 | 41.29 | 39.99 | 71.50 | 53.72 | 26.15 | 88.68 | 23.86 | 3.53 | 50.61 | 24.96 | 40.38 | 88.50 | 24.26 | 33.54 |
| **Quantizer Errors Handling** | | | | | | | | | | | | | | | | | |
| Quantizer-specific | 2.16 | 7.03 | 44.26 | 42.42 | 39.98 | 71.5 | 52.36 | 25.65 | 88.67 | 24.31 | 4.50 | 52.22 | 26.42 | 39.82 | 95.00 | 24.73 | 32.02 |
| Quantizer-agnostic | 2.16 | 7.13 | 44.26 | 42.95 | 41.11 | 71.5 | 53.46 | 25.99 | 88.32 | 24.03 | 4.50 | 50.60 | 25.80 | 39.05 | 95.50 | 24.41 | 32.43 |

# E. Detailed Evaluation of AQUA-KV and Baselines for Section 4.2

This appendix presents the evaluation results for AQUA-KV with the HIGGS backbone and baseline models across multiple architectures, including **Llama 3.2 (3B, 8B, 70B)** and **Qwen2.5 (3B, 7B)**. The models are assessed under **2-bit, 3-bit, and 4-bit KV-cache compression** settings.

We evaluate two key aspects of model performance:

*Table 10.* Evaluation of AQUA-KV (with HIGGS backbone) and baselines on Llama 3.2 3B for 2, 3 & 4 bit compression. The WikiText-2 Perplexity is evaluated on base (non-instruct) version of the model with sequence length 8192. The LongBench results are an average over 14 tasks evaluated with Instruct model with sequence length $2^{17}$ (131K) tokens. In addition to the overall average score across all tasks, the table also includes individual scores for each of the 14 tasks. The specific tasks in the original LongBench benchmark and the corresponding evaluation metrics can be found in the text.

| Config | Quant. Bits | Wiki2 PPL↓ (base model) | LongBench Avg ↑ (instruct model) | SamSum | 2WikiMQ | TREC | HotpotQA | MultiNews | TriviaQA | QMSum | PsgCount | MFQA_en | Musique | Qasper | PsgRetr | NarrativeQA | GovReport |
|---|---|---|---|---|---|---|---|---|---|---|---|---|---|---|---|---|---|
| Uncompressed | 16 | 6.98 | 44.61 | 42.50 | 40.32 | 70.50 | 52.77 | 25.79 | 88.78 | 24.38 | 5.00 | 51.13 | 26.21 | 40.74 | 97.00 | 24.93 | 34.54 |
| AQUA-KV | 2.09 | 7.03 | 44.30 | 41.76 | 40.66 | 72.50 | 52.72 | 25.46 | 88.78 | 24.43 | 4.50 | 49.07 | 26.21 | 39.36 | 97.00 | 25.58 | 32.13 |
| HIGGS | 2.02 | 7.47 | 42.80 | 39.56 | 39.97 | 72.5 | 52.54 | 24.52 | 87.76 | 24.10 | 3.00 | 49.61 | 26.91 | 36.84 | 88.50 | 25.35 | 28.00 |
| KIVI | 2.25 | 9.34 | 39.64 | 41.05 | 37.27 | 70.00 | 47.54 | 26.24 | 88.73 | 23.37 | 7.00 | 47.69 | 21.26 | 35.26 | 59.50 | 21.16 | 28.86 |
| KVQuant-2b-s1% | 2.33 | 9.43 | 18.28 | 26.75 | 17.38 | 50.50 | 23.58 | 21.64 | 56.96 | 18.28 | 2.11 | 22.93 | 8.25 | 12.92 | 4.50 | 4.41 | 17.66 |
| AQUA-KV | 3.06 | 6.98 | 44.37 | 43.16 | 40.67 | 72.5 | 52.19 | 25.77 | 88.39 | 24.71 | 4.00 | 49.25 | 26.55 | 39.47 | 96.50 | 24.43 | 33.52 |
| HIGGS | 3.02 | 7.05 | 44.41 | 42.74 | 40.67 | 73.00 | 52.54 | 25.58 | 88.80 | 24.70 | 4.50 | 48.96 | 26.21 | 40.11 | 96.50 | 24.61 | 32.87 |
| KIVI | 3.05 | 7.87 | 41.40 | 41.86 | 38.47 | 71.00 | 48.63 | 26.59 | 89.13 | 23.60 | 3.50 | 48.10 | 20.48 | 36.78 | 73.00 | 24.84 | 33.57 |
| KVQuant-3b-s1% | 3.33 | 7.26 | 23.85 | 42.17 | 35.40 | 72.00 | 46.75 | 25.51 | 89.05 | 23.85 | 4.53 | 50.79 | 22.54 | 41.43 | 70.50 | 24.03 | 31.03 |
| AQUA-KV | 4.02 | 6.98 | 44.48 | 42.86 | 40.59 | 72.5 | 52.18 | 25.69 | 88.78 | 24.36 | 4.50 | 49.28 | 26.06 | 40.30 | 96.50 | 25.20 | 33.89 |
| HIGGS | 4.02 | 7.01 | 44.41 | 42.06 | 40.91 | 72.50 | 52.40 | 25.63 | 88.22 | 24.55 | 4.00 | 49.77 | 26.89 | 40.31 | 96.00 | 24.70 | 33.74 |
| KIVI | 4.25 | 7.03 | 43.11 | 42.82 | 38.41 | 71.00 | 48.91 | 26.64 | 89.28 | 23.61 | 6.50 | 50.53 | 21.51 | 40.62 | 86.00 | 24.08 | 33.65 |
| KVQuant-4b-s1% | 4.33 | 7.04 | 24.20 | 42.44 | 37.14 | 72.50 | 50.84 | 25.84 | 88.44 | 24.20 | 2.00 | 53.12 | 23.06 | 38.99 | 94.50 | 24.52 | 33.05 |

*Table 11.* Evaluation of AQUA-KV (with HIGGS backbone) and baselines on Llama 3.1 8B for 2, 3 & 4 bit compression. The WikiText-2 Perplexity is evaluated on base (non-instruct) version of the model with sequence length 8192. The LongBench results are an average over 14 tasks evaluated with Instruct model with sequence length $2^{17}$ (131K) tokens. In addition to the overall average score across all tasks, the table also includes individual scores for each of the 14 tasks. The specific tasks in the original LongBench benchmark and the corresponding evaluation metrics can be found in the text.

| Config | Quant. Bits | Wiki2 PPL↓ (base model) | LongBench Avg ↑ (instruct model) | SamSum | 2WikiMQ | TREC | HotpotQA | MultiNews | TriviaQA | QMSum | PsgCount | MFQA_en | Musique | Qasper | PsgRetr | NarrativeQA | GovReport |
|---|---|---|---|---|---|---|---|---|---|---|---|---|---|---|---|---|---|
| Uncompressed | 16 | 5.61 | 48.13 | 43.62 | 48.58 | 72.50 | 57.8 | 26.86 | 91.47 | 25.43 | 10.50 | 55.58 | 32.75 | 44.62 | 100.0 | 29.65 | 34.4 |
| AQUA-KV | 2.08 | 5.72 | 47.77 | 42.99 | 48.16 | 73.50 | 57.58 | 26.15 | 91.91 | 25.79 | 7.25 | 55.71 | 33.46 | 44.53 | 99.50 | 29.67 | 32.53 |
| HIGGS | 2.02 | 5.89 | 47.37 | 41.20 | 49.03 | 73.50 | 57.47 | 25.21 | 91.97 | 25.21 | 7.25 | 56.91 | 31.80 | 43.76 | 99.5 | 30.27 | 30.16 |
| KIVI | 2.25 | 7.37 | 46.28 | 43.41 | 43.33 | 71.50 | 55.12 | 26.79 | 91.14 | 24.57 | 5.67 | 52.56 | 30.9 | 41.02 | 99.50 | 29.10 | 33.3 |
| KVQuant-2b-s1% | 2.33 | 6.64 | 37.17 | 41.94 | 34.23 | 65.00 | 43.78 | 25.82 | 86.27 | 22.94 | 2.92 | 47.13 | 23.24 | 36.51 | 36.50 | 22.63 | 31.53 |
| AQUA-KV | 3.05 | 5.64 | 48.10 | 43.89 | 48.35 | 73.5 | 57.43 | 26.85 | 91.48 | 25.46 | 7.43 | 56.64 | 33.96 | 45.18 | 99.5 | 29.61 | 34.06 |
| HIGGS | 3.02 | 5.66 | 47.86 | 43.75 | 47.77 | 73.5 | 57.74 | 26.41 | 91.93 | 25.39 | 7.33 | 55.67 | 33.57 | 43.97 | 99.5 | 29.87 | 33.6 |
| KIVI | 3.05 | 6.04 | 46.98 | 43.46 | 43.86 | 72.50 | 54.46 | 26.93 | 91.76 | 25.21 | 6.78 | 54.40 | 30.92 | 43.73 | 99.00 | 29.54 | 35.15 |
| KVQuant-3b-s1% | 3.33 | 5.84 | 46.42 | 44.56 | 42.73 | 72.50 | 53.91 | 26.54 | 92.01 | 24.96 | 5.42 | 53.42 | 28.03 | 44.31 | 99.50 | 28.64 | 33.40 |
| AQUA-KV | 4.02 | 5.61 | 48.10 | 44.14 | 48.81 | 73.5 | 57.2 | 27.11 | 91.63 | 25.47 | 7.43 | 56.17 | 33.69 | 44.6 | 99.5 | 29.68 | 34.45 |
| HIGGS | 4.02 | 5.62 | 48.07 | 43.6 | 48.82 | 73.5 | 57.23 | 26.89 | 91.47 | 25.42 | 7.6 | 55.72 | 34.29 | 44.43 | 99.5 | 30.28 | 34.21 |
| KIVI | 4.25 | 5.64 | 47.57 | 44.07 | 45.54 | 72.50 | 55.72 | 26.86 | 92.42 | 25.58 | 8.34 | 54.89 | 31.41 | 44.42 | 99.50 | 29.46 | 35.22 |
| KVQuant-4b-s1% | 4.33 | 5.65 | 47.77 | 43.61 | 49.10 | 71.00 | 58.56 | 26.87 | 91.03 | 25.25 | 5.42 | 55.03 | 33.41 | 45.67 | 99.50 | 29.86 | 34.45 |

1. **Language Modeling Quality** – Measured using *WikiText-2 Perplexity*, tested on the **base (non-instruct)** versions of the models with a sequence length of 8192.

2. **Task-Specific Performance** – Measured on *LongBench*, a benchmark covering **14 diverse NLP tasks**, with evaluation conducted on the **Instruct-tuned** versions of the models using a sequence length of $2^{17}$ (131K tokens) in total.

The Tables 10-14 summarize the average LongBench performance across all 14 tasks for each model, followed by the WikiText-2 perplexity scores. Additionally, detailed per-task results are provided for a deeper understanding of how different models and quantization settings affect specific NLP capabilities.

To further analyze the impact of KV-cache compression on various NLP applications, we report the performance on each of the **14 individual LongBench tasks** in the following table 10. Each task has its own original name, evaluation metric and average length, reflecting its unique requirements.

*Table 12.* Evaluation of AQUA-KV (with HIGGS backbone) and baselines on Llama 3.1 70B for 2, 3 & 4 bit compression. The WikiText-2 Perplexity is evaluated on base (non-instruct) version of the model with sequence length 8192. The LongBench results are an average over 14 tasks evaluated with Instruct model with sequence length $2^{17}$ (131K) tokens. In addition to the overall average score across all tasks, the table also includes individual scores for each of the 14 tasks. The specific tasks in the original LongBench benchmark and the corresponding evaluation metrics can be found in the text.

| Config | Quant. Bits | Wiki2 PPL↓ (base model) | LongBench Avg ↑ (instruct model) | SamSum | 2WikiMQ | TREC | HotpotQA | MultiNews | TriviaQA | QMSum | PsgCount | MFQA_en | Musique | Qasper | PsgRetr | NarrativeQA | GovReport |
|---|---|---|---|---|---|---|---|---|---|---|---|---|---|---|---|---|---|
| Uncompressed | 16 | 2.54 | 52.92 | 47.04 | 68.63 | 76.50 | 65.71 | 26.72 | 94.21 | 23.93 | 18.00 | 54.79 | 45.96 | 49.73 | 98.50 | 36.31 | 34.85 |
| AQUA-KV | 2.04 | 2.62 | 52.79 | 46.65 | 69.71 | 76.00 | 64.94 | 26.64 | 94.04 | 24.40 | 18.00 | 55.31 | 46.17 | 49.72 | 97.50 | 35.92 | 34.06 |
| HIGGS | 2.02 | 2.77 | 52.18 | 46.43 | 68.04 | 76.00 | 63.41 | 26.49 | 93.94 | 24.25 | 20.00 | 54.08 | 44.56 | 48.68 | 97.50 | 35.04 | 32.11 |
| KIVI | 2.25 | 3.06 | 52.45 | 47.07 | 66.83 | 76.50 | 63.88 | 26.43 | 93.38 | 24.50 | 20.00 | 53.91 | 46.66 | 49.19 | 97.50 | 34.89 | 33.58 |
| KVQuant-2b-s1% | 2.33 | 3.28 | 31.39 | 44.87 | 39.64 | 65.50 | 51.52 | 26.09 | 89.42 | 23.44 | 11.50 | 51.64 | 38.50 | 43.59 | 97.50 | 30.92 | 31.85 |
| AQUA-KV | 3.03 | 2.55 | 52.81 | 46.74 | 68.63 | 76.50 | 65.11 | 26.50 | 94.21 | 24.16 | 18.00 | 54.60 | 45.89 | 49.73 | 98.50 | 36.17 | 34.59 |
| HIGGS | 3.02 | 2.57 | 52.56 | 46.26 | 68.33 | 75.50 | 64.76 | 26.38 | 94.04 | 24.80 | 18.50 | 54.74 | 45.17 | 49.24 | 97.50 | 36.42 | 34.24 |
| KIVI | 3.05 | 2.87 | 52.87 | 47.66 | 67.09 | 76.50 | 64.37 | 26.66 | 93.61 | 24.56 | 20.00 | 54.54 | 47.56 | 49.87 | 98.00 | 35.47 | 34.29 |
| KVQuant-3b-s1% | 3.33 | 2.75 | 35.02 | 45.83 | 59.45 | 76.00 | 58.21 | 26.32 | 89.42 | 23.90 | 17.50 | 53.86 | 43.93 | 48.46 | 97.50 | 35.86 | 34.18 |
| AQUA-KV | 4.02 | 2.54 | 52.95 | 47.05 | 68.63 | 76.50 | 65.79 | 26.75 | 94.04 | 24.01 | 18.50 | 54.66 | 46.18 | 49.49 | 98.50 | 36.58 | 34.63 |
| HIGGS | 4.02 | 2.55 | 52.97 | 47.77 | 69.01 | 76.50 | 66.33 | 26.83 | 94.04 | 24.50 | 16.50 | 54.37 | 46.56 | 49.27 | 99.00 | 36.31 | 34.65 |
| KIVI | 4.25 | 2.61 | 52.88 | 47.24 | 67.27 | 76.50 | 64.37 | 26.75 | 94.04 | 24.47 | 20.00 | 54.01 | 47.85 | 50.19 | 97.50 | 35.23 | 34.91 |
| KVQuant-4b-s1% | 4.33 | 2.58 | 35.42 | 47.68 | 68.79 | 75.50 | 66.70 | 26.87 | 93.73 | 24.20 | 18.00 | 54.27 | 46.03 | 48.83 | 99.00 | 36.13 | 34.70 |

*Table 13.* Evaluation of AQUA-KV (with HIGGS backbone) and baselines on Qwen2.5 3B for 2, 3 & 4 bit compression. The WikiText-2 Perplexity is evaluated on base (non-instruct) version of the model with sequence length 8192. The LongBench results are an average over 14 tasks evaluated with Instruct model with sequence length $2^{17}$ (131K) tokens. In addition to the overall average score across all tasks, the table also includes individual scores for each of the 14 tasks. The specific tasks in the LongBench benchmark and the corresponding evaluation metrics can be found in the text. The increased AQUA-KV bitwidth is due to not quantizing the 1st block for this model.

| Config | Quant. Bits | Wiki2 PPL↓ (base model) | LongBench Avg ↑ (instruct model) | SamSum | 2WikiMQ | TREC | HotpotQA | MultiNews | TriviaQA | QMSum | PsgCount | MFQA_en | Musique | Qasper | PsgRetr | NarrativeQA | GovReport |
|---|---|---|---|---|---|---|---|---|---|---|---|---|---|---|---|---|---|
| Uncompressed | 16 | 7.14 | 38.80 | 44.45 | 38.64 | 68.00 | 46.60 | 22.60 | 87.60 | 22.94 | 3.00 | 49.29 | 20.53 | 37.07 | 49.00 | 21.88 | 31.64 |
| AQUA-KV | 2.44 | 7.20 | 38.31 | 43.43 | 37.43 | 68.50 | 46.35 | 22.69 | 88.17 | 23.23 | 3.00 | 48.93 | 18.11 | 36.98 | 47.50 | 21.71 | 30.34 |
| HIGGS | 2.06 | 7.92 | 30.92 | 38.73 | 34.81 | 48.00 | 44.54 | 18.11 | 84.40 | 20.26 | 2.50 | 38.24 | 16.17 | 32.51 | 15.00 | 19.34 | 20.26 |
| KIVI | 2.25 | 9.05 | 28.66 | 42.07 | 12.48 | 69.00 | 18.60 | 23.37 | 87.05 | 24.09 | 5.05 | 29.96 | 10.56 | 11.10 | 31.27 | 10.31 | 26.31 |
| AQUA-KV | 3.42 | 7.15 | 38.77 | 45.19 | 38.64 | 68.00 | 46.81 | 22.90 | 87.67 | 23.35 | 2.50 | 48.43 | 20.39 | 37.34 | 47.50 | 22.79 | 31.29 |
| HIGGS | 3.06 | 7.28 | 31.85 | 42.90 | 30.92 | 53.00 | 41.33 | 21.64 | 79.05 | 22.53 | 2.50 | 42.94 | 15.98 | 29.94 | 20.50 | 19.78 | 22.94 |
| KIVI | 3.05 | 7.63 | 30.37 | 44.08 | 13.71 | 69.00 | 18.31 | 23.97 | 86.66 | 23.44 | 4.75 | 36.62 | 9.84 | 14.10 | 38.75 | 12.05 | 29.92 |
| AQUA-KV | 4.4 | 7.14 | 38.92 | 44.77 | 37.98 | 68.00 | 46.80 | 22.57 | 87.93 | 23.09 | 3.00 | 49.07 | 21.20 | 37.34 | 50.00 | 21.75 | 31.42 |
| HIGGS | 4.06 | 7.15 | 32.11 | 41.37 | 29.80 | 55.00 | 37.84 | 23.01 | 70.86 | 22.79 | 2.00 | 42.52 | 15.46 | 34.17 | 33.42 | 20.29 | 20.95 |
| KIVI | 4.25 | 7.17 | 31.50 | 45.01 | 15.42 | 69.50 | 21.40 | 24.59 | 87.53 | 23.76 | 3.50 | 38.99 | 12.06 | 16.13 | 42.50 | 8.65 | 31.9 |

*Table 14.* Evaluation of AQUA-KV (with HIGGS backbone) and baselines on Qwen2.5 7B for 2, 3 & 4 bit compression. The WikiText-2 Perplexity is evaluated on base (non-instruct) version of the model with sequence length 8192. The LongBench results are an average over 14 tasks evaluated with Instruct model with sequence length $2^{17}$ (131K) tokens. In addition to the overall average score across all tasks, the table also includes individual scores for each of the 14 tasks. The specific tasks in the LongBench benchmark and the corresponding evaluation metrics can be found in the text. The increased AQUA-KV bitwidth is due to not quantizing the 1st block for this model.

| Config | Quant. Bits | Wiki2 PPL↓ (base model) | LongBench Avg ↑ (instruct model) | SamSum | 2WikiMQ | TREC | HotpotQA | MultiNews | TriviaQA | QMSum | PsgCount | MFQA_en | Musique | Qasper | PsgRetr | NarrativeQA | GovReport |
|---|---|---|---|---|---|---|---|---|---|---|---|---|---|---|---|---|---|
| Uncompressed | 16 | 6.13 | 46.82 | 45.77 | 46.94 | 72.00 | 57.72 | 23.90 | 89.42 | 23.56 | 8.00 | 52.58 | 30.35 | 43.78 | 100.00 | 29.49 | 31.93 |
| AQUA-KV | 2.48 | 6.17 | 46.43 | 45.99 | 45.71 | 72.00 | 56.88 | 23.84 | 89.18 | 23.44 | 8.50 | 52.15 | 29.63 | 42.77 | 99.50 | 29.21 | 31.15 |
| HIGGS | 2.03 | 8.08 | 25.97 | 26.98 | 17.98 | 52.50 | 31.67 | 12.04 | 55.46 | 13.14 | 7.25 | 27.19 | 14.55 | 22.22 | 55.79 | 14.72 | 12.05 |
| KIVI | 2.25 | 7.02 | 32.78 | 44.79 | 10.60 | 71.00 | 11.51 | 22.04 | 86.71 | 21.25 | 6.41 | 27.29 | 6.69 | 12.81 | 91.67 | 14.37 | 31.79 |
| AQUA-KV | 3.45 | 6.14 | 46.81 | 45.58 | 47.35 | 72.00 | 57.88 | 24.07 | 89.10 | 23.59 | 8.00 | 52.63 | 30.64 | 44.05 | 100.00 | 28.40 | 32.03 |
| HIGGS | 3.03 | 7.2 | 14.61 | 14.02 | 13.38 | 45.00 | 15.27 | 6.07 | 29.29 | 10.00 | 5.54 | 18.29 | 4.65 | 14.81 | 15.90 | 6.10 | 6.19 |
| KIVI | 3.05 | 6.37 | 32.63 | 45.65 | 9.71 | 71.00 | 10.34 | 22.49 | 88.87 | 20.68 | 4.69 | 29.83 | 6.87 | 13.17 | 90.46 | 10.66 | 32.37 |
| AQUA-KV | 4.43 | 6.14 | 46.77 | 45.81 | 46.87 | 72.00 | 57.71 | 24.06 | 89.83 | 23.83 | 8.00 | 52.41 | 30.35 | 43.37 | 100.00 | 28.65 | 31.86 |
| HIGGS | 4.03 | 6.88 | 11.54 | 11.80 | 7.62 | 37.75 | 10.13 | 4.39 | 22.54 | 10.52 | 7.20 | 10.50 | 4.83 | 5.85 | 16.26 | 7.11 | 5.02 |
| KIVI | 4.25 | 6.14 | 33.40 | 46.20 | 9.61 | 71.00 | 10.34 | 22.27 | 89.63 | 21.13 | 4.58 | 30.56 | 6.85 | 12.79 | 98.25 | 12.06 | 32.39 |

*Table 15.* LongBench tasks mapping and evaluation metrics.

| Our task name | Orig. task name | Eval metric | Avg len |
|---|---|---|---|
| SamSum | SAMSum | Rouge-L | 6,258 |
| 2WikiMQ | 2WikiMultihopQA | F1 | 4,887 |
| TREC | TREC | Accuracy | 5,177 |
| HotpotQA | HotpotQA | F1 | 9,151 |
| MultiNews | MultiNews | Rouge-L | 2,113 |
| TriviaQA | TriviaQA | F1 | 8,209 |
| QMSum | QMSum | Rouge-L | 10,614 |
| PsgCount | PassageCount | Accuracy | 11,141 |
| MFQA_en | MutiFieldQA-en | F1 | 4,559 |
| Musique | MUSiQue | F1 | 11,214 |
| Qasper | Qasper | F1 | 3,619 |
| PsgRetr | PassageRetrieval-en | Accuracy | 9,289 |
| NarrativeQA | NarrativeQA | F1 | 18,409 |
| GovReport | GovReport | Rouge-L | 8,734 |

# F. Additional Evaluations and Details for Section 4.3

In Section 4.3, we evaluate the following benchmark settings:

1. GSM8K (Cobbe et al., 2021) mathematical reasoning in 8-shot chain-of-thought setup.

2. MMLU-Pro (Wang et al., 2024) general reasoning in 5-shot chain-of-thought setup.

3. IFEval (Zhou et al., 2023) instruction following benchmark in zero-shot mode.

We use Language Model Evaluation Harness (Gao et al., 2021) version 0.4.7 with default parameters for each benchmark. For MMLU-Pro, Table 6 in the main paper reports aggregated accuracy over 14 different subsets. For completeness, we also report the individual per-task scores in Table 16.

*Table 16.* Evaluation of AQUA-KV (with HIGGS backbone) and baselines on Llama 3.2 3B Instruct and Llama 3.1 8B Instruct with 2 bit compression. In addition to the overall average MMLU-Pro score, the table also includes individual scores for each of the 14 tasks

| Config | Quant. Bits | Overall score (instruct model) | Biology | Business | Chemistry | CompSci | Economics | Engineering | Health | History | Law | Math | Other | Philosophy | Physics | Psychology |
|---|---|---|---|---|---|---|---|---|---|---|---|---|---|---|---|---|
| | | | | | | | **Llama 3.2 3B** | | | | | | | | | |
| Uncompressed | 16 | 34.47 | 53.14 | 35.61 | 26.68 | 37.07 | 42.65 | 21.98 | 40.59 | 32.81 | 23.52 | 35.23 | 35.61 | 33.07 | 28.56 | 50.25 |
| AQUA-KV | 2.09 | 32.79 | 51.60 | 32.19 | 23.76 | 34.15 | 41.71 | 21.26 | 38.02 | 34.12 | 23.43 | 32.86 | 33.33 | 34.67 | 25.25 | 50.38 |
| HIGGS | 2.02 | 29.25 | 47.56 | 28.64 | 21.91 | 29.51 | 40.52 | 18.99 | 31.05 | 29.40 | 18.26 | 26.94 | 31.60 | 32.46 | 22.02 | 48.37 |
| | | | | | | | **Llama 3.1 8B** | | | | | | | | | |
| Uncompressed | 16 | 44.34 | 64.99 | 48.04 | 35.51 | 49.76 | 55.57 | 30.24 | 51.59 | 42.52 | 27.70 | 44.86 | 45.45 | 44.69 | 39.03 | 59.77 |
| AQUA-KV | 2.08 | 43.25 | 61.92 | 46.89 | 32.95 | 46.10 | 55.57 | 28.79 | 50.73 | 42.26 | 28.52 | 42.04 | 45.56 | 44.29 | 37.95 | 61.03 |
| HIGGS | 2.02 | 38.40 | 59.41 | 38.53 | 28.71 | 36.34 | 50.71 | 25.39 | 43.89 | 38.58 | 28.34 | 33.23 | 42.86 | 42.08 | 31.18 | 58.15 |

Additionally, we compare AQUA-KV and HIGGS compression on the HumanEval (Chen et al., 2021) programming benchmark using the default LM Evaluaton Harness configuration. The results are summarized in Table 17.

While AQUA-KV similarly outperforms quantization without predictors, we caution the reader against drawing significant conclusions from these HumanEval results. The absolute scores, even without quantization, do not match the publicly reported HumanEval scores for Llama 3.x. Upon further inspection, we found that this is a common issue and other researchers also report problems[4] when reproducing HumanEval. We are uncertain whether this is due to Instruct

---

[4]E.g. see https://github.com/meta-llama/llama3/issues/101

*Table 17.* HumanEval pass@1 results

| Method | Quant. Bits | Llama 3.2 3B | Qwen 2.5 3B |
|---|---|---|---|
| Uncompressed | 16 | 24.28 | 17.38 |
| AQUA-KV | 2.09 | 24.25 | 17.31 |
| HIGGS | 2.02 | 22.26 | 17.18 |

model prompting, answer extraction or other issues. The problem may potentially be resolved in the recently introduced `humaneval_instruct` task in LM Evaluation Harness that was added after we ran our experiments, but we did not have the compute budget to test it further.

## G. Additional Evaluations of AQUA-KV with $H_2O$

Here, we report more detailed evaluation results in a setup where AQUA-KV is combined with $H_2O$ Heavy Hitter Oracle. In addition to per-task LongBench scores, we also report several additional configurations for AQUA-KV backbone quantizer. The results for 3B and 8B models can be found in Table 20 and 19 respectively.

*Table 18.* Evaluation of Llama 3.2 3B Instruct with $H_2O$ pruning mixed with various Key-Value cache compression strategies. A 20% KV cache budget for $H_2O$ was used for all evaluations. The left panel contains the average LongBench score for the model. The right panel reports detailed per-task LongBench accuracies and F1 scores for the model.

| Config | Quant. Bits | LongBench Avg ↑ (instruct model) | SamSum | 2WikiMQ | TREC | HotpotQA | MultiNews | TriviaQA | QMSum | PsgCount | MFQA_en | Musique | Qasper | PsgRetr | NarrativeQA | GovReport |
|---|---|---|---|---|---|---|---|---|---|---|---|---|---|---|---|---|
| Uncompressed | 16 | 44.61 | 42.5 | 40.32 | 70.5 | 52.77 | 25.79 | 88.78 | 24.38 | 5 | 51.13 | 26.21 | 40.74 | 97 | 24.93 | 34.54 |
| $H_2O$ | 16 | 38.82 | 42.92 | 39.87 | 68 | 43.91 | 22.64 | 88.88 | 22.14 | 7.5 | 48.52 | 16.38 | 31.78 | 65 | 18.39 | 27.54 |
| $H_2O$ + AQUA-KV | 2.09 | 38.43 | 42.99 | 39.67 | 68 | 44.02 | 22.19 | 88.6 | 21.74 | 7.5 | 46.51 | 16.54 | 30.03 | 65 | 18.65 | 26.52 |
| $H_2O$ + HIGGS | 2.02 | 37.02 | 40.04 | 37.93 | 67.5 | 44.11 | 22.15 | 87.45 | 21.19 | 6.5 | 46.5 | 16.05 | 29.07 | 56.5 | 18.07 | 25.24 |
| $H_2O$ + AQUA-KV | 3.06 | 38.76 | 43.2 | 39.73 | 68 | 43.99 | 22.34 | 89.04 | 21.93 | 7.5 | 48.02 | 16.51 | 31.5 | 64.5 | 18.88 | 27.45 |
| $H_2O$ + HIGGS | 3.02 | 38.27 | 42.96 | 37.81 | 67.5 | 43.3 | 22.15 | 89.38 | 22.11 | 8.5 | 46.56 | 16.35 | 30.17 | 62.5 | 19.34 | 27.14 |
| $H_2O$ + AQUA-KV | 4.02 | 38.85 | 43.16 | 39.96 | 68 | 43.9 | 22.86 | 89.04 | 22.17 | 7.5 | 48.44 | 16.38 | 31.88 | 64.5 | 18.48 | 27.61 |
| $H_2O$ + HIGGS | 4.02 | 38.74 | 42.66 | 38.19 | 68 | 43.91 | 22.31 | 88.78 | 22.03 | 7 | 48.59 | 17.06 | 32.01 | 65.5 | 18.92 | 27.34 |

*Table 19.* Evaluation of Llama 3.1 8B Instruct with $H_2O$ pruning mixed with various Key-Value cache compression strategies. A 20% KV cache budget for $H_2O$ was used for all evaluations. The left panel contains the average LongBench score for the model. The right panel reports detailed per-task LongBench accuracies and F1 scores for the model.

| Config | Quant. Bits | LongBench Avg ↑ (instruct model) | SamSum | 2WikiMQ | TREC | HotpotQA | MultiNews | TriviaQA | QMSum | PsgCount | MFQA_en | Musique | Qasper | PsgRetr | NarrativeQA | GovReport |
|---|---|---|---|---|---|---|---|---|---|---|---|---|---|---|---|---|
| Uncompressed | 16 | 48.13 | 43.62 | 48.58 | 72.5 | 57.8 | 26.86 | 91.47 | 25.43 | 10.5 | 55.58 | 32.75 | 44.62 | 100 | 29.65 | 34.4 |
| $H_2O$ | 16 | 41.42 | 44.57 | 44.84 | 69.5 | 44.09 | 23.04 | 91.75 | 22.1 | 6.61 | 51.49 | 24.28 | 38.45 | 68 | 22.29 | 28.91 |
| $H_2O$ + AQUA-KV | 2.09 | 41.11 | 44.35 | 42.57 | 69 | 44.3 | 22.6 | 92.19 | 22.06 | 6.59 | 51.87 | 23.92 | 37.57 | 68.5 | 21.9 | 28.16 |
| $H_2O$ + HIGGS | 2.02 | 40.72 | 42.04 | 44 | 68.5 | 43.39 | 23.01 | 91.26 | 22.38 | 7.24 | 50.08 | 24.24 | 34.53 | 69 | 22.68 | 27.69 |
| $H_2O$ + AQUA-KV | 3.06 | 41.31 | 44.63 | 44.69 | 69.5 | 44.32 | 23.06 | 92.28 | 22.12 | 6.51 | 50.68 | 23.98 | 37.45 | 68 | 22.65 | 28.4 |
| $H_2O$ + HIGGS | 3.02 | 41.37 | 44.39 | 45.14 | 69.5 | 44.88 | 22.86 | 91.84 | 22.5 | 6.95 | 50.72 | 24.24 | 37.27 | 68 | 22.57 | 28.31 |
| $H_2O$ + AQUA-KV | 4.02 | 41.47 | 44.51 | 44.94 | 69.5 | 44.07 | 22.95 | 91.75 | 22.45 | 6.83 | 51.53 | 24.29 | 38.5 | 68 | 22.22 | 28.98 |
| $H_2O$ + HIGGS | 4.02 | 41.32 | 44.47 | 44.08 | 69.5 | 43.97 | 23.03 | 91.93 | 22.11 | 6.83 | 50.52 | 24.34 | 38.67 | 68 | 22.21 | 28.8 |

## H. Detailed Inference Benchmarks

In this section, we report detailed inference speed benchmarks using our AQUA-KV implementation. Our implementation consists of two CUDA kernels:

1. **Quantization kernel:** finds the nearest vector in a HIGGS lattice and returns its index. Since the original HIGGS (Mali-

*Table 20.* Evaluation of Llama 3.2 3B Instruct with $H_2O$ pruning mixed with various Key-Value cache compression strategies. A 50% KV cache budget for $H_2O$ was used for all evaluations. The left panel contains the average LongBench score for the model. The right panel reports detailed per-task LongBench accuracies and F1 scores for the model.

| Config | Quant. Bits | LongBench Avg ↑ (instruct model) | SamSum | 2WikiMQ | TREC | HotpotQA | MultiNews | TriviaQA | QMSum | PsgCount | MFQA.en | Musique | Qasper | PsgRetr | NarrativeQA | GovReport |
|---|---|---|---|---|---|---|---|---|---|---|---|---|---|---|---|---|
| Uncompressed | 16 | 44.61 | 42.5 | 40.32 | 70.5 | 52.77 | 25.79 | 88.78 | 24.38 | 5 | 51.13 | 26.21 | 40.74 | 97 | 24.93 | 34.54 |
| $H_2O$ | 16 | 39.69 | 42.84 | 37.87 | 68.5 | 44.57 | 24.84 | 88.22 | 22.51 | 8.5 | 48.87 | 16.72 | 37.22 | 65 | 19.35 | 30.59 |
| $H_2O$ + AQUA-KV | 2.09 | 39.11 | 42.89 | 38.04 | 68.5 | 45.12 | 24.38 | 88.64 | 21.82 | 6 | 49.15 | 16.59 | 34.65 | 63.5 | 19.68 | 28.62 |
| $H_2O$ + HIGGS | 2.02 | 37.42 | 39.72 | 36.61 | 68.5 | 44.01 | 24.15 | 87.52 | 21.77 | 5 | 47.9 | 16.01 | 32.32 | 54.5 | 18.85 | 27.07 |
| $H_2O$ + AQUA-KV | 3.06 | 39.61 | 42.59 | 38.68 | 68.5 | 44.58 | 25.06 | 88.39 | 22.17 | 7.5 | 49.22 | 17.22 | 36.46 | 64.5 | 19.42 | 30.29 |
| $H_2O$ + HIGGS | 3.02 | 39.21 | 42.38 | 37.62 | 68 | 45.04 | 24.75 | 89.06 | 22.32 | 9 | 47.96 | 16.14 | 36.21 | 61 | 19.5 | 29.93 |
| $H_2O$ + AQUA-KV | 4.02 | 39.72 | 42.65 | 37.8 | 68.5 | 44.58 | 25.07 | 89.08 | 22.35 | 8.5 | 49.06 | 17.06 | 36.83 | 64 | 20.07 | 30.52 |
| $H_2O$ + HIGGS | 4.02 | 39.45 | 42.23 | 38.1 | 68.5 | 44.67 | 25 | 88.53 | 22.37 | 9 | 49.78 | 16.16 | 36.85 | 61 | 19.81 | 30.3 |

*Table 21.* Evaluation of Llama 3.1 8B Instruct with $H_2O$ pruning mixed with various Key-Value cache compression strategies. A 50% KV cache budget for $H_2O$ was used for all evaluations. The left panel contains the average LongBench score for the model. The right panel reports detailed per-task LongBench accuracies and F1 scores for the model.

| Config | Quant. Bits | LongBench Avg ↑ (instruct model) | SamSum | 2WikiMQ | TREC | HotpotQA | MultiNews | TriviaQA | QMSum | PsgCount | MFQA.en | Musique | Qasper | PsgRetr | NarrativeQA | GovReport |
|---|---|---|---|---|---|---|---|---|---|---|---|---|---|---|---|---|
| Uncompressed | 16 | 48.13 | 43.62 | 48.58 | 72.5 | 57.8 | 26.86 | 91.47 | 25.43 | 10.5 | 55.58 | 32.75 | 44.62 | 100 | 29.65 | 34.4 |
| $H_2O$ | 16 | 42.35 | 44.36 | 44.46 | 69.5 | 43.68 | 25.53 | 91.83 | 23.15 | 6.49 | 52.93 | 24.8 | 43.63 | 68 | 22.78 | 31.79 |
| $H_2O$ + AQUA-KV | 2.09 | 42.08 | 43.87 | 43.32 | 69 | 44.27 | 25.31 | 92.38 | 23.45 | 6.14 | 53.2 | 24.86 | 41.8 | 68.5 | 22.7 | 30.31 |
| $H_2O$ + HIGGS | 2.02 | 41.61 | 41.69 | 44.87 | 69.5 | 43.55 | 25.33 | 91.52 | 22.47 | 6.55 | 52.43 | 24.08 | 41.11 | 68 | 22.24 | 29.13 |
| $H_2O$ + AQUA-KV | 3.05 | 42.37 | 44.1 | 44.65 | 69.5 | 44.12 | 25.38 | 92.62 | 23.47 | 6.5 | 53.57 | 24.73 | 43.3 | 68 | 22.13 | 31.14 |
| $H_2O$ + HIGGS | 3.02 | 42.23 | 43.84 | 45.28 | 69.5 | 43.94 | 25.34 | 91.85 | 23.07 | 6.49 | 53.16 | 24.85 | 42.01 | 68 | 22.41 | 31.45 |
| $H_2O$ + AQUA-KV | 4.02 | 42.37 | 44.3 | 44.34 | 69.5 | 43.73 | 25.78 | 91.72 | 23.33 | 6.71 | 53.08 | 24.7 | 43.56 | 68 | 22.82 | 31.64 |
| $H_2O$ + HIGGS | 4.02 | 42.17 | 44.38 | 43.65 | 69.5 | 43.54 | 25.81 | 91.93 | 22.84 | 6.76 | 53.25 | 24.54 | 42.53 | 68 | 22.13 | 31.58 |

novskii et al., 2024b) did not implement fast quantization, we developed this kernel specifically for our use case.

2. **Dequantization kernel:** selects the corresponding vector from the lattice, applies quantization scale and performs Hadamard transformation. The dequantized values are added to the predictor outputs in the same kernel.

We compare three inference modes: standard (bfloat16) inference, quantized AQUA-KV inference with CUDA kernels, and quantized inference with naive PyTorch (Paszke et al., 2019) implementation. The models we benchmark on are Llama 3.2 3B and Llama 3.1 70B to test our approach with different model sizes.

**Inference latency.** Our first set of experiments measures inference latency with batch size 1. We generate sequences of up to length 32768 and report the full forward pass latency (time per token). The results in Figure 6 show that our implementation introduces an average inference overhead of 60% for shorter sequences (fewer than 10k tokens), and an average overhead of 18% for longer sequences (10k to 32k tokens). This is expected since AQUA-KV performs all the computations required for bfloat16 inference and additional quantization/dequantization operations.

**Throughput.** Next, we compare infernce throughput in a batched inference setting. For each method, we select the maximum batch size that fits on GPU (A100 for 3B, $2\times$ A100 for 70B) at maximum sequence length and measure the number of tokens generated per second. In this setting, AQUA-KV compressed can achieve greater throughput for shorter sequences as shown in Figure 7. Note, however, that this improvement stems from the fact that our method can fit a larger batch size in the same GPU memory due to cache compression.

Overall, our CUDA implementation significantly outperforms the naive Python / PyTorch inference due to having significantly less DRAM I/O operations (from fused dequantization). The latency overhead can be further reduced with additional optimizations: for instance, we can quantize the predictors themselves to 4-bit with almost no loss in accuracy (see Appendix D), which could allow us to speed up predictor computations. Additionally, it should be possible to fuse quantized cache reconstruction with the attention computation for further speedups.

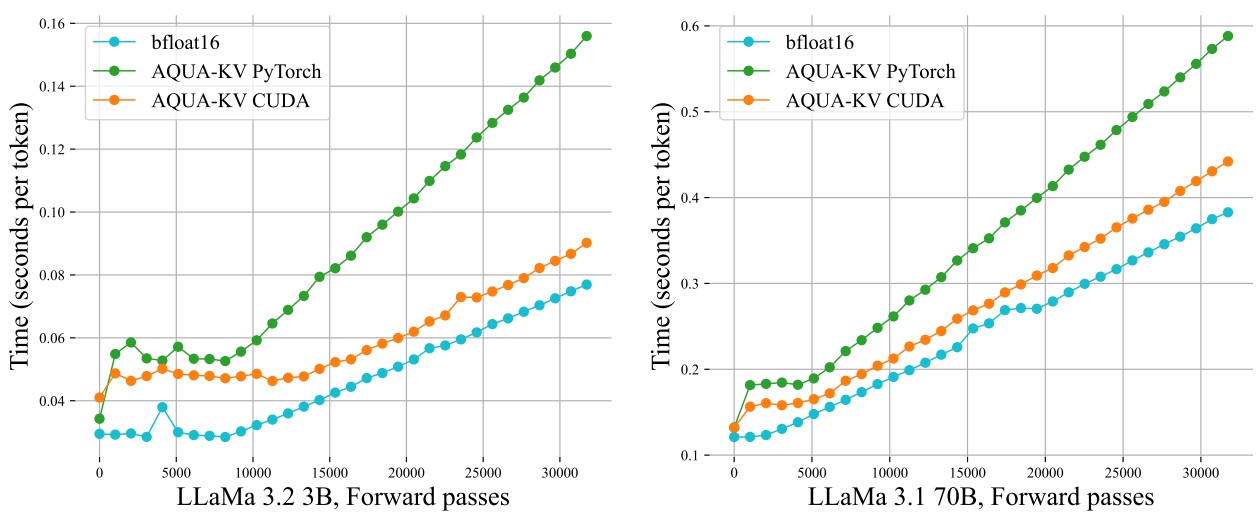

*Figure 6.* Inference latency with batch size 1 for Llama 3.2 3B (left) and Llama 3.1 70B (right) on A100 GPUs.

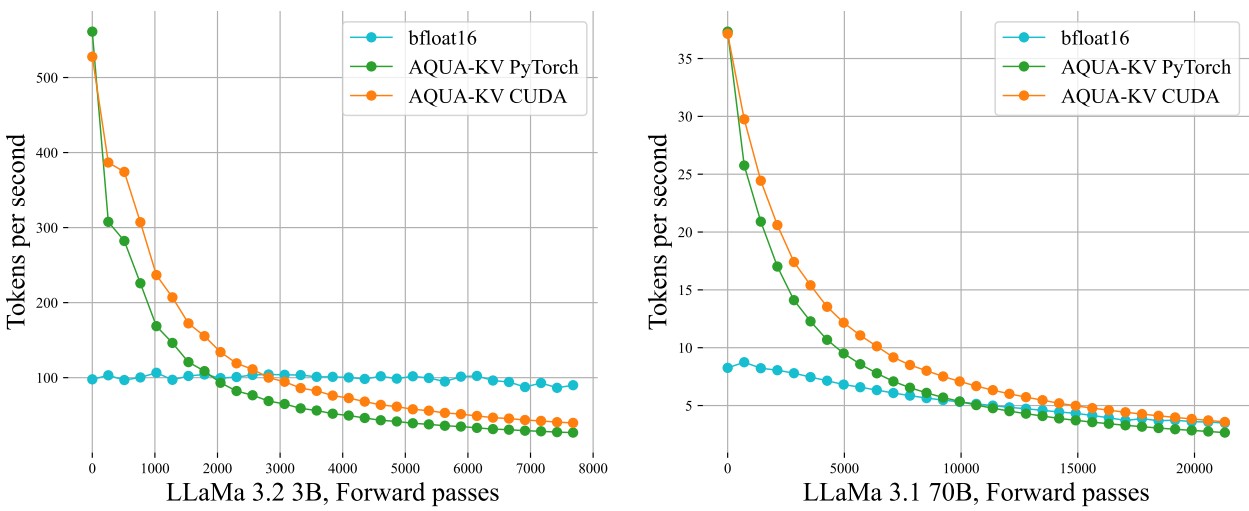

*Figure 7.* Throughput of batched inference for Llama 3.2 3B (left) and Llama 3.1 70B (right) on A100 GPUs.

