# OpenReview forum: "Cache Me If You Must: Adaptive Key-Value Quantization for Large Language Models"
_ICML.cc/2025/Conference — ICML 2025 poster_

### Official Review · Reviewer_MQGD · 2025-03-01

**Overall Recommendation:** 3

**Summary:**

The paper proposes AQUA-KV, a KV cache compression method for autoregressive LLMs that explots inter and intra layer dependencies for improving cache quantization accuracy. It can be combined with additional compression techniques such as pruning. To this effect, they train predictors for predicting the value of a Key & Value pair using other cache entries, and apply quantization on the residual information that could not be predicted. This method only needs to store the information that cannot be recovered from other sources. They use the previous layer keys to predict the subsequent keys, and use both the previous layer values and the current layer keys to predict values in their implementation. In their practical implementation they use linear regression for all predictors, and any quantization method can be used for quantizing the residual information. For evaluation they measure perplexity and end to end performance. They measure perplexity on WikiText-2 and end to end performance accuracy on LongBench tasks.

**Claims And Evidence:**

I am mildly concerned about conclusions drawn from Table 2 based on the end to end performance of their method and other methods on LongBench. I highlight this concern and phrase it as a question to the authors below.

**Essential References Not Discussed:**

I am not aware of any important references not discussed.

**Experimental Designs Or Analyses:**

Yes I checked the evaluation of the proposed method with various other cache quantization schemes for 2 bit quantization, when evaluated on perplexity and end to end accuracy on LongBench. I also checked the evaluation of the proposed method with using HIGGS as the quantization mechanism as compared to other quantization methods for 5 different LLMs on the perplexity and end to end performance tasks.

**Methods And Evaluation Criteria:**

Yes.

**Other Comments Or Suggestions:**

None.

**Other Strengths And Weaknesses:**

I think the main strength of the proposed method is that it is modular and can be easily combined with any quantization method, and it can be applied on top of any token pruning method. This really increases the versatility of the procedure.

**Questions For Authors:**

My main concern is that in Table 2, where the authors compare their method to other quantization approaches across various LLMs, their method shows a 0.3 improvement over HIGGS on the LongBench average score for Llama 3.X 8B and 70B in the 3-bit quantization setting but requires an additional 0.5 GB of cache memory. Similarly, for 2-bit quantization, it achieves a 0.4-0.6 improvement but demands 0.6 GB more memory on the LongBench average score for Llama 3.X 8B and 70B. This suggests that the experiment may not be fully controlled, making it unclear whether the observed gains in LongBench performance are significant or if any improvement would persist under a similar cache size constraint. Can the authors have a more controlled experiment, or explain why the existing experiment is controlled enough ?

The other question I have is whether the authors compared with KV cache compression techniques based on token selection and pruning, such as StreamingLLM, SnapKV since the authors have compared their method with H2O.

**Relation To Broader Scientific Literature:**

The main contribution of the paper is to the literature on KV cache compression and quantization techniques for memory efficient inference in LLMs.

**Theoretical Claims:**

The paper does not have any theoretical claims.

---

> ### Author Rebuttal · Authors · 2025-03-31
>
> We thank the reviewer for their feedback and suggestions. We appreciate that you highlight the modularity of AQUA-KV and address your concerns below.
>
> > My main concern is that in Table 2, where the authors compare their method to other quantization approaches across various LLMs, their method shows a 0.3 improvement over HIGGS on the LongBench average score for Llama 3.X 8B and 70B in the 3-bit quantization setting but requires an additional 0.5 GB of cache memory. Similarly, for 2-bit quantization, it achieves a 0.4-0.6 improvement but demands 0.6 GB more memory on the LongBench average score for Llama 3.X 8B and 70B. This suggests that the experiment may not be fully controlled, making it unclear whether the observed gains in LongBench performance are significant or if any improvement would persist under a similar cache size constraint. Can the authors have a more controlled experiment, or explain why the existing experiment is controlled enough ?
>
> We agree that our comparison can be improved by controlling strictly for the cache size. To address this, we conducted additional evaluations with raw HIGGS (our strongest baseline), where the quantizer was given more quantization clusters to increase the average bitwidth. To recall, a 2.x bit HIGGS splits the data into $d{=}2$-dimensional vectors and rounds each vector to one of $n{=}2^4{=}16$ clusters, with an additional 16-bit scale per $g{=}1024$ quantized values (the default configuration from [1]). This yields $\log_2 n / d + 16/g \approx 2.0156$ bits per parameter.
>
> [1] https://arxiv.org/abs/2411.17525
>
> To offset AQUA-KV predictors, we evaluate HIGGS with a lattice of $n{=}18$ clusters instead of 16, with an average bitwidth of $\approx 2.1$, resulting in a slightly larger overall cache size than for AQUA-KV.
>
> ## Table: WikiText-2 Perplexity (non-Instruct models, setup from Section 4.2)
>
> | Method | Avg. Bits | Llama 3.2 3B | 3.1 8B |
> |--------------------|----------|--------------|--------|
> | - | 16 | 6.98 | 5.61 |
> | AQUA-KV | 2.09 |**7.03** | **5.72** |
> | HIGGS ($n=16$) | 2.02 | 7.47 | 5.89 |
> | HIGGS ($n=18$) | 2.10 | 7.40 | 5.85 |
>
> ## Table: Average LongBench scores (Instruct models, setup from Section 4.2)
>
> | Method | Avg. Bits | Llama 3.2 3B | 3.1 8B |
> |--------------------|----------|--------------|--------|
> | - | 16 | 44.61 | 48.13 |
> | AQUA-KV | 2.09 | **44.30** | **47.77** |
> | HIGGS ($n=16$) | 2.02 | 42.80 | 47.37 |
> | HIGGS ($n=18$) | 2.10 | 43.31 | 47.16 |
>
> As we can see, HIGGS with additional clusters does indeed perform better at the cost of a greater memory footprint, but AQUA-KV still outperforms it.
>
>
> Please also note that the AQUA-KV memory overhead can be reduced by quantizing the predictor weights. We report this setup in Table 1 (see “GPTQ”, L359 for 4-bit quantization). We discuss this in L346-358 (right) and report more detailed results in Table 4 (Appendix).
>
> We hope that these new results alleviate the reviewer’s concern and will add them to Section 4.1 and perform additional evaluations in Appendix.
>
> > The other question I have is whether the authors compared with KV cache compression techniques based on token selection and pruning, such as StreamingLLM, SnapKV since the authors have compared their method with H2O.
>
> In our work, we chose the token pruning proposed in the H$_2$O paper as it was a middle ground between StreamingLLM [1] and more recent methods such as SnapKV [2]. In principle, AQUA-KV can also be combined with other token pruning strategies, including the two you proposed. We agree that exploring these combinations can further strengthen our paper, but we need additional time to incorporate them into our codebase and ensure that our experiment setup uses these pruning strategies properly. We thank the reviewer for the suggestion and will add these comparisons in Section 4.3 in the final version of the paper.
>
>
> [1] https://arxiv.org/abs/2309.17453 Xiao et al, 2023. Efficient Streaming Language Models with Attention Sinks
>
>
> [2] https://arxiv.org/abs/2404.14469 Li et al, 2024, SnapKV: LLM Knows What You are Looking for Before Generation

---

> > ### Comment · Reviewer_MQGD · 2025-04-07
> >
> > Thanks the detailed responses and clarifications, I will keep my current evaluation.

---

### Official Review · Reviewer_LU4T · 2025-03-09

**Overall Recommendation:** 3

**Summary:**

This work proposes AQUA-KV, a method of using inter-layer and intra-layer information to reduce the size of the KV cache with minimum overhead via a supplementary probe. AQUA-KV supplements a “backbone” quantization algorithm, where it functions to improve accuracy by using the information available from the previous layer (for both the keys and values) and within the same layer (for the values). By this method, only residual information unique to each token need be saved in the KV cache, allowing for better compression.

**Claims And Evidence:**

The claim that AQUA-KV contributes more to accuracy than the baseline quantization is somewhat supported, especially for smaller models. However, there are too few evaluations of larger models (Llama 70B) to be certain that the proposed method adds value.

**Essential References Not Discussed:**

None

**Experimental Designs Or Analyses:**

The experimental design of checking results for different backbone quantization algorithms and probes was sound. Also, the authors provided sufficient analysis of the effects of using multiple previous layers and previous tokens. The major issue was the paucity of evaluations.

**Methods And Evaluation Criteria:**

The limited number and types of evaluations conducted are the main weakness in this work. Although long sequence evaluation is also important, evaluations on shorter sequences should also be conducted. For example, using batched inference could also necessitate KV cache compression.

Moreover, in Table 2 of the paper, there is only a small difference between the performance of HIGGS on Llama 3.1 70B compared to AQUA-KV with HIGGS. Much more rigorous evaluation is required to see if AQUA-KV is effective for large models as well as smaller models. Language generation tasks by instruction-tuned models should be evaluated rigorously as these are closest to those used for actual LLM production. MMLU, GSM8K, HumanEval, and IFEval are frequently used.

I am willing to change my rating if these concerns are addressed.

**Other Comments Or Suggestions:**

There appears to be a missing word in paragraph 3 of Section 4. “We evaluate perplexity on base (non-instruct) models since they have better .”

The title displayed on the first page does not match the title displayed on the top of the subsequent pages.

**Other Strengths And Weaknesses:**

The method is simple to understand and has relatively little overhead, both during the training and inference stages. Although the authors do not provide an optimized implementation, there does not appear to be any fundamental barrier to integrating their solution to frameworks such as vLLM.

Also, by demonstrating the effectiveness of their method on the LongBench, the authors show that their method is competitive with other KV cache quantization algorithms on long-sequence evaluations, even matching the performance of the unquantized BF16 baselines.

**Questions For Authors:**

Could the prediction for the keys be overlapped with the calculation of the previous FFN layer? This could reduce the overhead from the probe, although some KV cache quantization methods may not be compatible.

Could the benefit of applying the keys to reconstruct the values be investigated more thoroughly? This creates a dependency that prevents overlapping.

**Relation To Broader Scientific Literature:**

KV cache quantization is an emerging research topic with high practical value. With the increasing volume of LLM inference, reducing KV cache memory is a key consideration for LLM service providers. The authors propose a new method of improving outcomes from compressing the KV cache while retaining accuracy.

**Theoretical Claims:**

There were no theoretical claims in this work.

---

> ### Author Rebuttal · Authors · 2025-03-31
>
> We thank the reviewer for their valuable feedback. Overall, the review appreciates the efficacy and simplicity of AQUA-KV, but suggests additional evaluations on extra benchmarks and asks follow-up questions about implementation details. We do our best to address these below.
>
> # Additional evaluations
>
> > In Table 2 of the paper, there is only a small difference between the performance of HIGGS on Llama 3.1 70B compared to AQUA-KV with HIGGS. Much more rigorous evaluation is required to see if AQUA-KV is effective for large models as well as smaller models.
>
> First, we would like to note that the quantized 70B model already has very high quality, meaning that any absolute changes in score will be small.
> Thus, we ask the reviewer to take into account the *relative* difference: for 2.x bits per value, the 70B model perplexity with 16-bit cache is 2.54. AQUA-KV @ 2-bit increases only PPL by +0.08, whereas raw HIGGS increases it by +0.23 (*almost 3x*). Likewise, the LongBench score drops by 0.13 for AQUA-KV and 0.74 for the nearest 2-bit baseline (>5x error increase).
>
> To fully address the remark concerning larger-scale evaluations, we evaluate the 72B Qwen 2.5 model in the same setup as in Section 4.2:
>
> |Model|Method|Avg.Bits|Wiki2PPL (non-Instruct)|GSM8K (Instruct)|
> |-|-|-|-|-|
> |72B|-|16|3.49|95.8|
> |72B|AQUA-KV|2.09|**3.56**|**95.5**|
> |72B|HIGGS|2.02|3.66|93.7|
>
> We also report additional benchmarks for Llama 3.1 70B below and in our response to Reviewer AgDD. To further explore the scalability, we will run the remaining evaluations for 70B+ models in the final version of the paper.
>
> > Language generation tasks by instruction-tuned models should be evaluated rigorously as these are closest to those used for actual LLM production. MMLU, GSM8K, HumanEval, and IFEval are frequently used.
>
> We agree and evaluate GSM8K and IFEVAL across different models, including 70B. We prioritize 2.x bit evaluations due to time constraints and since this is where augmenting quantizers makes the most sense.
>
>
> **GSM8K accuracy (%) for Instruct models in the same setup as Section 4.2**
>
> |Method|Avg.Bits|Llama 3.2 3B|3.1 8B|3.1 70B|Qwen 2.5 3B| 7B|
> |-|-|-|-|-|-|-|
> |Uncompressed|16|76.5|85.1|94.7|61.2|76.6|
> |AQUA-KV|2.09|**77.7**|**84.3**|**94.2**|**59.9**|**72.2**|
> |HIGGS|2.02|70.3|79.2|**94.2**|35.8|59.7|
>
> **IFEval accuracy (%) for Instruct models in the same setup as Section 4.2**
>
> |Method|Avg.Bits|Llama 3.2 3B|3.1 8B|3.1 70B|Qwen 2.5 3B| 7B|
> |-|-|-|-|-|-|-|
> |Uncompressed|16|77.0|78.9|88.0|66.5|76.9|
> |AQUA-KV|2.09|**75.1**|**79.9**|**88.1**|**66.2**|66.9|
> |HIGGS|2.02|72.4|75.7|87.0|59.3|**68.6**|
>
> The results show a similar trend to our LongBench evaluations, with AQUA-KV being substantially closer to the uncompressed baseline than raw HIGGS.
> We will include these results in the final version of the paper and conduct additional experiments with the other two benchmarks (MMLU and HumanEval).
>
> > Although long sequence evaluation is also important, evaluations on shorter sequences should also be conducted.
>
> We hope that this concern can be alleviated with the results we reported above, since those benchmarks have shorter sequences (e.g. GSM8K question plus answer takes up, on average, **198 tokens** for Llama-3.1/3.2 tokenizer). We also report perplexity with shorter sequence length (see the first table in our response to Reviewer AgDD). We do note, however, that KV-cache compression is most effective in the long-context regime.
>
> # Questions about overlapping AQUA-KV with model inference
>
> > Could the prediction for the keys be overlapped with the calculation of the previous FFN layer?
>
> Thank you for this suggestion. It is indeed possible to overlap FFN/MLP computation with computing the next layer AQUA-KV cache. Furthermore, **since the value predictor is linear, we can overlap half of its computation (from previous values) as well,** then add the other half after the fact.
>
> > Could the benefit of applying the keys to reconstruct the values be investigated more thoroughly? This creates a dependency that prevents overlapping.
>
> We have investigated this question via ablation analysis in **Table 4, section “predictor inputs”, on L809 (w/o $K_{rec}$ → V)**. To summarize, the key predictor does improve perplexity and LongBench scores, but only slightly (in the last digit). This component can indeed be removed in cases where one cares about better overlap. We will discuss this trade-off in Section 4.1.
>
> > I am willing to change my rating if these concerns are addressed.
>
> We hope that the additional evaluations and discussions we provided can alleviate the reviewer’s concerns. If you add any follow-up suggestions in the next discussion phase, we will address them in the final version of the paper.

---

### Official Review · Reviewer_8Gvk · 2025-03-13

**Overall Recommendation:** 3

**Summary:**

The paper presents AQUA-KV, an approach that leverages dependencies between keys and values across adjacent attention blocks. The method employs linear predictors trained to estimate KV caches for a given block based on previously generated keys and values. Subsequently, the residuals are quantized to low bit-widths to achieve efficient KV cache compression.

## update after rebuttal
The paper presents a novel approach to reduce the KV cache quantization errors. After rebuttal, the authors clarify several parts in the paper and add more evaluations. Therefore, I recommend to accept this paper.

**Claims And Evidence:**

Yes.

**Essential References Not Discussed:**

None.

**Experimental Designs Or Analyses:**

Yes.

**Methods And Evaluation Criteria:**

Yes.

**Other Comments Or Suggestions:**

N/A

**Other Strengths And Weaknesses:**

Strengths: The idea is simple and effective, and it can quantize the stored KV cache residuals to 2-bit.

Weaknesses:
1. The paper is somehow difficult to follow. See Questions below.
2. The experiments are only conducted on Wiki2PPL and some LongBench tasks. It lacks evaluations on more challenging tasks such as math and code tasks with CoT prompts.

**Questions For Authors:**

1. What does "high-compression mechanisms for internal network states" mean in the abstract? Is it an observation?
2. In L59-60 in the paper, the author mentioned "vector quantization". However, I can not find any details.
3. L196-197 needs to be elaborated.

**Relation To Broader Scientific Literature:**

N/A

**Theoretical Claims:**

N/A

---

> ### Author Rebuttal · Authors · 2025-03-31
>
> We thank the reviewer for the thoughtful feedback and valuable suggestions. We agree that improving clarity and expanding evaluations would strengthen the paper, and we address these points below:
>
> > What does "high-compression mechanisms for internal network states" mean in the abstract? Is it an observation?
>
> We meant to say that there exist methods that quantize KV states to low-bitwidth with small quality degradation. In the final version of the paper it could be rephrased to “In this work, we aim to improve Key & Value compression by exploiting two observations: … 2) the existence of high-compression methods for internal network states (e.g. attention Keys & Values).”
>
> > In L59-60 in the paper, the author mentioned "vector quantization". However, I can not find any details.
>
> By “vector quantization” we referred to HIGGS, which is a vector quantization method, as described in L161. We meant that using HIGGS with predictors provides the best quality compared to other methods for KV quantization, as discussed further in Table 2.
> We will clarify that by adding “and the more advanced **vector quantization scheme** HIGGS” in L272.
>
> > L196-197 needs to be elaborated.
>
> These lines are indeed somewhat convoluted. We meant the following:
> 1. we noticed that using 1- and 2-bit quantizers (e.g. HIGGS) can ‘explain’ ~0.75 and ~0.89 of the variance respectively. In other words, they have a ~0.25 and ~0.11 relative quantization error.
> 2. If a probe can predict keys/values with the same error as a 1-bit quantizer, we found that we can use 1 less bit for quantization (e.g. 3-bit instead of 4-bit) after the residual with, on average, the same accuracy (e.g. see Table 2).
>
> We will clarify this in the revised paper.
>
> > The experiments are only conducted on Wiki2PPL and some LongBench tasks. It lacks evaluations on more challenging tasks such as math and code tasks with CoT prompts.
>
> We agree that AQUA-KV can benefit from additional evaluations on larger models. As requested, we have conducted evaluations on GSM8k (CoT) and IFEval and report them to in the tables below.
>
> **GSM8k accuracy (%) for Instruct models in the same setup as Section 4.2**
>
> |Method|Avg.Bits|Llama 3.2 3B|3.1 8B|3.1 70B|Qwen 2.5 3B| 7B|
> |-|-|-|-|-|-|-|
> |Uncompressed|16|76.5|85.1|94.7|61.2|76.6|
> |AQUA-KV|2.09|**77.7**|**84.3**|**94.2**|**59.9**|**72.2**|
> |HIGGS|2.02|70.3|79.2|**94.2**|35.8|59.7|
>
> **IFEval accuracy (%) for Instruct models in the same setup as Section 4.2**
>
> |Method|Avg.Bits|Llama 3.2 3B|3.1 8B|3.1 70B|Qwen 2.5 3B| 7B|
> |-|-|-|-|-|-|-|
> |Uncompressed|16|77.0|78.9|88.0|66.5|76.9|
> |AQUA-KV|2.09|**75.1**|**79.9**|**88.1**|**66.2**|66.9|
> |HIGGS|2.02|72.4|75.7|87.0|59.3|**68.6**|
>
> The results generally align with the trends observed in LongBench evaluations, with AQUA-KV being substantially closer to the uncompressed baseline than raw HIGGS. We will add these evaluations to the final version of the paper and conduct additional experiments for code benchmarks (e.g. HumanEval).
>
> We appreciate the reviewer’s insightful comments, which have helped us improve the paper’s clarity and experimental scope. The additional evaluations confirm AQUA-KV’s consistent performance across domains, as noted in our response. We hope these revisions address all raised concerns. If you add any follow-up suggestions in the next discussion phase, we will address them in the final version of the paper.

---

> > ### Comment · Reviewer_8Gvk · 2025-04-08
> >
> > Thanks to the authors for the response. I will keep my score.

---

### Official Review · Reviewer_AgDD · 2025-03-18

**Overall Recommendation:** 4

**Summary:**

The paper proposes a learned predictor based adaptive quantization for KV Cache compression. The idea is this -- transformer models are residual in nature, i.e. each subsequent layers add smaller and smaller deltas to the outputs -- this means that the intermediate representations are highly dependent. This turns out to be true for even KV Cache vectors. Paper leverages this understanding to train simple linear predictors for KV Cache and only quantize the residuals. since liner predictors explain significant variance the quantization on residuals are correspondingly accurate leading to further compression.

**Claims And Evidence:**

Yes. The claims are well supported

**Essential References Not Discussed:**

I am not well versed with related literature

**Experimental Designs Or Analyses:**

Experimental setup seems correct. Longbenchmark is a representative benchmark for KV Cache issue. The bit compressions used are reasonable. The baselines seem reasonable. ( Disclaimer: I am not very well versed with baselines in this field. For instance, are there any other adaptive methods that can be combined with quantization / pruning -- since no such method is discussed in the paper)

**Methods And Evaluation Criteria:**

Yes. The method is novel and well-suited application of micro-machine learning.

**Other Comments Or Suggestions:**

None.

**Other Strengths And Weaknesses:**

[Strengths]
1. Good use of learned predictors
2. Good gains in memory footprints at same accuracy
3. The impact on efficiency in contained.
4. Empirical evaluation including ablations is useful.

[Weakness]
Nothing I can think of.

**Questions For Authors:**

[Questions out of curiosity , do not impact the evaluation of the paper]

1. If you train your predictors post-ROPE for 8192 sequences, do you see any deterioration in inference on sequences beyond 8192 tokens?
2. How do you think about combining multiple compression techniques together -- low-rank, quantization, predictor-based, pruning,etc.
3. What in your opinion is the lower-bound (bits / token) on compression for reasonable performance. Like can we go below 2 bits?

**Relation To Broader Scientific Literature:**

The idea of using local learned components in compression is a new idea, in my understanding. Post-training compression is generally fixed algorithms. Learning is used in compression in two ways -- intermittent training of entire model for recovering compression loss (QAT or LTH) or from-scratch training of compressed models (SynFLOW like pruning OR ROAST). The idea of using compact ML models for reducing dimension / variance of data while natural is new in compression literature.

**Theoretical Claims:**

No theoretical claims made

---

> ### Author Rebuttal · Authors · 2025-03-31
>
> We thank the reviewer for their thoughtful feedback and are glad that they appreciate our method's design and empirical results.
> Below, we provide detailed answers to the questions posed in the review:
>
> > If you train your predictors post-ROPE for 8192 sequences, do you see any deterioration in inference on sequences beyond 8192 tokens?
>
> In short, we found that AQUA-KV is not sensitive to the training sequence length: specifically, we did not see a significant impact on longer sequences present in our LongBench evaluation (with some tasks in excess of 100k tokens [1]). We attribute this to the fact that AQUA-KV only trains simple linear predictors.
>
> [1] https://github.com/THUDM/LongBench
>
> However, we agree that it is important to analyze the effect of training sequence length. To that end, we evaluate AQUA-KV for the Llama 3.2 3B model with **varying training sequence length** and measure the impact on perplexity.
>
> Table 1. WikiText-2 PPL for different training sequence lengths.
>
> |Eval \ Train sequence length|128|1024|4096|8192|
> |-|-|-|-|-|
> |8192  |7.02|7.03|7.03|7.03|
> |128  |17.87|17.87|17.87|17.87|
>
> Note that in these evaluations, we control for the total number of tokens: every time we halve the sequence length, we also double the number of sequences in the calibration dataset. Otherwise, training with 64-token sequences would overfit because of insufficient training dataset size. For convenience, we report additional sequence lengths in https://anonymous.4open.science/r/rebuttal-pics-24A3/.
>
> > How do you think about combining multiple compression techniques together -- low-rank, quantization, predictor-based, pruning,etc.
>
> Our approach is indeed compatible with different cache compression techniques applied simultaneously. Specifically, **we combined AQUA-KV with H$_2$O pruning in Section 4.3 (detailed results provided in Appendix E)** and show that AQUA-KV can augment H$_2$O to further improve its size-to-accuracy trade-offs by combining three techniques together: pruning (H$_2$O), predictors (AQUA-KV) and quantization (HIGGS). We also explored low-rank *predictors* in Appendix C (Table 4), which can further reduce the memory footprint at the cost of degraded performance. In case you are interested in other specific combinations, we will consider them and add to the final version of the paper.
>
>
>
>
>
> > What in your opinion is the lower-bound (bits / token) on compression for reasonable performance. Like can we go below 2 bits?
>
> In our work, we focused on ~2 bits per value because this setup can achieve favorable quality-to-size trade-offs for practitioners. However, it is indeed interesting to evaluate AQUA-KV in the extreme sub 2-bit setup. To that end, we evaluate AQUA-KV on top of a 1-bit HIGGS variant with d=8 group dimension and n=256 clusters, with the rest of hyperparameters matching our setup from Section 4.1. This results in circa 1 bit per stored value. We report our results for Llama 3.2 3B and 3.1 8B below.
>
> Table 2. WikiText-2 perplexity evaluation of AQUA-KV for 1 bit compression.
> | Method | Avg. Bits | Llama 3.2 3B | 3.1 8B |
> |--------------------|----------|--------------|--------|
> | - | 16 | 6.98 | 5.61 |
> | AQUA-KV ($d{=}8$, $n{=}256$) | 1.09 | **7.52** | **6.10** |
> | HIGGS ($d{=}8$, $n{=}256$) | 1.02 | 16.18 | 19.83 |
>
> Table 3. LongBench evaluation of AQUA-KV for 1 bit compression.
>
> | Method | Avg. Bits | Llama 3.2 3B (Instruct) | 3.1 8B (Instruct) |
> |--------------------|----------|--------------|--------|
> | - | 16 | 44.61 | 48.13 |
> | AQUA-KV ($d{=}8$, $n{=}256$) | 2.09 | **40.61** | **43.09** |
> | HIGGS ($d=8$, $n{=}256$) | 2.02 | 23.02 | 24.94 |
>
> To summarize, AQUA-KV with 1-bit HIGGS quantization can achieve substantially better quality than raw 1-bit quantization, but both methods show higher quality deterioration. Still, this is an interesting setup and a potential frontier for future research. We will add these and additional sub 2-bit evaluations to the final version of the paper.

---

> > ### Comment · Reviewer_AgDD · 2025-04-02
> >
> > Thanks for the response. I will maintain my recommendation to accept the paper.

---

### Decision · Program_Chairs · 2025-05-01

**Decision:**

Accept (poster)

**Comment:**

This paper presents AQUA-KV, an effective method for adaptive KV cache quantization in LLMs. It introduces compact linear predictors to exploit inter- and intra-layer dependencies in the KV cache, enabling residual quantization that achieves higher compression rates with minimal loss in model quality. The idea is both simple and insightful, with strong empirical results across multiple model sizes and benchmarks. The authors also proactively addressed reviewers' concerns by providing additional evaluations on large-scale models and by comparing controlled memory setups, showing consistent gains over strong baselines. While some reviewers initially raised concerns about experimental scope and clarity, the rebuttal convincingly responds to them. Given the thorough empirical validation, I recommend acceptance.